# Fleet of Agents: Coordinated Problem Solving with Large Language Models

Lars Klein [* 1]   Nearchos Potamitis [* 2]   Roland Aydin [3]   Robert West [1]   Caglar Gulcehre [1]   Akhil Arora [2]

## Abstract

While numerous frameworks have been developed to enhance the reasoning abilities of large language models (LLMs), there is a scarcity of methods that effectively balance the trade-off between cost and quality. In this paper, we introduce FLEET OF AGENTS (FOA), a novel and intuitive yet principled framework utilizing LLMs as agents to navigate through dynamic tree searches, employing a *genetic-type particle filtering* approach. FOA spawns a multitude of agents, each exploring the search space autonomously, followed by a *selection phase* where resampling based on a heuristic value function optimizes the balance between exploration and exploitation. This mechanism enables *dynamic branching*, adapting the exploration strategy based on discovered solutions. We conduct extensive experiments on four benchmark tasks, "Game of 24", "Mini-Crosswords", "WebShop" and "SciBench", utilizing four different LLMs, GPT-3.5, GPT-4, LLaMA3.2-11B, and LLaMA3.2-90B. On average across all tasks and LLMs, FOA obtains an *absolute quality improvement of* $\simeq 5\%$ while requiring *only* $\simeq 35\%$ *of the cost* of previous SOTA methods. Notably, our analyses reveal that (1) FOA achieves the best cost-quality trade-off among all benchmarked methods, and (2) FOA + LLaMA3.2-11B surpasses the Llama3.2-90B model. FOA is publicly available at https://github.com/au-clan/FoA.

## 1. Introduction

With strong reasoning and problem-solving abilities, large language models (LLMs) such as GPT-4 (Achiam et al., 2024), LLaMA (Touvron et al., 2023a;b; Grattafiori et al.,

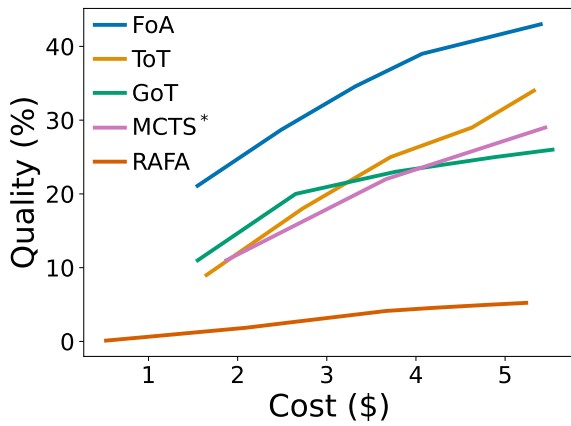

Figure 1: Analyzing the trade-off between cost and quality of representative SOTA methods with GPT-3.5 on the Game of 24 task. **FOA achieves the best cost-quality trade-off.**

2024), and PaLM (Anil et al., 2023), have sparked a newfound interest in building general-purpose autonomous agents. LLM-based agents have portrayed excellent performance on reasoning and knowledge-intensive tasks (Cobbe et al., 2021), often requiring interactions with complex environments, such as playing complex video games (Fan et al., 2022), performing web navigation (Yao et al., 2022), or enabling tool-use (Schick et al., 2023).

Naturally, the rise of LLM-based agents has contributed to the prosperity of prompt-based reasoning frameworks (Wei et al., 2022; Besta et al., 2024; Yang et al., 2024; Yao et al., 2024; Lingam et al., 2025; Shinn et al., 2023; Yao et al., 2023) that further enhance the problem-solving and reasoning abilities of LLMs. Broadly, the reasoning frameworks can be categorized into two categories: (1) single-query reasoning and (2) multi-query reasoning. As the name implies, single-query methods (Wei et al., 2022; Wang et al., 2023; Sel et al., 2024; Nye et al., 2021; Kojima et al., 2022) obtain an answer by querying the LLM only once, whereas, multi-query methods (Yao et al., 2024; Besta et al., 2024; Zhou et al., 2024; Shinn et al., 2023; Yao et al., 2023) perform multiple LLM queries to identify different plausible reasoning paths or to plan ahead. It is important to note that none of the two aforementioned paradigms is perfect.

On the one hand, despite being cost-effective by design, single-query methods require one or more of the following: intricate prompt engineering, high-quality demonstrations,

---

[*]Equal contribution; authors listed in alphabetical order. [1]EPFL [2]Aarhus University [3]TUHH. Correspondence to: Nearchos Potamitis <nearchos.potamitis@cs.au.dk>, Lars Klein <lars.klein@epfl.ch>, Akhil Arora <akhil.arora@cs.au.dk>.

*Proceedings of the 42nd International Conference on Machine Learning*, Vancouver, Canada. PMLR 267, 2025. Copyright 2025 by the author(s).

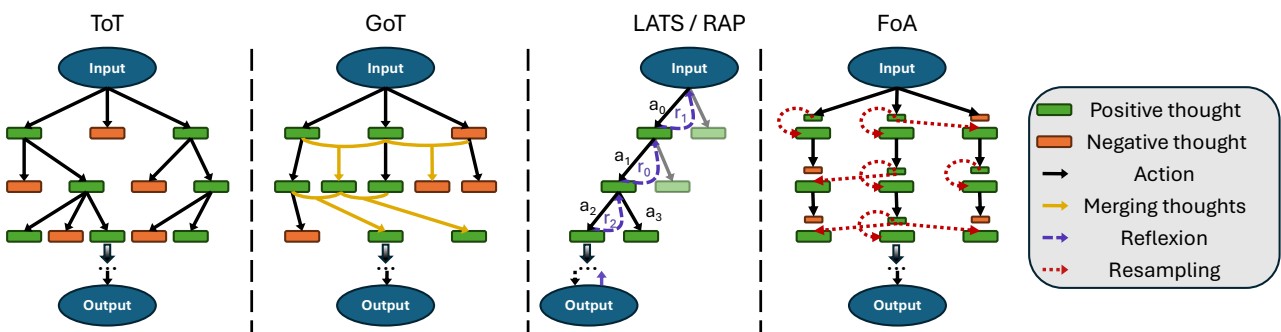

Figure 2: Comparison between SOTA tree-search-based reasoning (Yao et al., 2024; Besta et al., 2024; Zhou et al., 2024; Hao et al., 2023) and our FOA frameworks. FOA offers precise control over the tree width ($n$ agents) and depth ($t$ steps), leading to predictable latency and cost. However, by expanding the $c$ most promising states at each step, tree-search methods offer no such control and their search trees might grow exponentially.

or knowledge distilled from informative historical reasoning processes, to achieve competitive quality. More importantly, even then, these methods are not well-suited for sequential decision-making tasks that require interactions with an environment, such as web navigation (Yao et al., 2022).

On the other hand, multi-query methods decompose a complex problem into a series of simpler sub-problems and search over all plausible reasoning paths. This allows them to obtain competitive quality but also renders them inefficient and expensive. To devise a reasoning framework applicable to both general problem-solving and sequential decision-making tasks, our focus in this paper is on improving the cost-efficiency of multi-query methods.

**Present work.** We introduce FLEET OF AGENTS (FOA), a novel and intuitive yet principled framework that brings the concept of genetic-type particle filtering (Holland, 1992) to dynamic tree searches. Fig. 2 provides an overview of our framework, while at the same time highlighting the conceptual differences between state-of-the-art (SOTA) tree-search-based methods (Yao et al., 2024; Besta et al., 2024; Zhou et al., 2024; Hao et al., 2023) and FOA.

FOA spawns a multitude of agents, each exploring the search space autonomously, followed by a *selection phase* where *resampling* based on a heuristic value function optimizes the balance between exploration and exploitation. If an agent has discovered a promising solution approach indicated by a state with a high value, the resampling mechanism may create multiple copies of this agent. Conversely, if none of the agents is ahead, or in other words, there are multiple promising states, the resampling mechanism may retain all of them, thereby maintaining a fleet of high diversity. This mechanism enables *dynamic branching*, adapting the exploration strategy based on discovered solutions.

**Cost-quality trade-off.** The biggest advantage of FOA is its ability to strike a balance between exploration vs. ex-

ploitation. We provide early empirical evidence in Fig. 1, which compares the performance of tree-based SOTA methods with FOA for varying price points. We find that FOA substantially outperforms the existing SOTA methods at all possible price points, thereby achieving the best cost-quality trade-off among the benchmarked methods.

**Contributions.**

• We propose an intuitive yet principled framework, FLEET OF AGENTS (FOA), for improving the cost-quality trade-off of LLM-based reasoning (§ 3).

• Ours is the first work to explore the concept of genetic particle filtering in the context of AI agents (cf. § 2 for a detailed literature review).

• At its core, FOA is a runtime that can readily integrate any existing AI agent, without having to change their behavior. We compare FOA against various SOTA reasoners by incorporating their agents, unchanged, into our framework.

• We conduct extensive experiments on four benchmark tasks using four LLMs as base models. On average across all tasks and LLMs, **FOA obtains a quality improvement of $\simeq 5\%$ while requiring only $\simeq 35\%$ of the cost** of previous SOTA methods (§ 4 and § 5).

## 2. Related Work

In this section, we review works that overlap closely with our study (cf. Appx. A for additional related work).

**Prompt-based reasoning.** Recent research focuses on developing strategies to enhance the reasoning capabilities of LLMs. Few-shot prompting employs demonstrations of high-quality input/output samples to exploit the eagerness of the LLMs to imitate patterns seen in their context window (Brown et al., 2020). Algorithm of thoughts (AoT) (Sel et al., 2024), goes a step further by including algorithmic exam-

ples within the prompt to propel the LLM through algorithmic reasoning pathways. Chain-of-Thought (CoT) prompting (Nye et al., 2021; Wei et al., 2022; Kojima et al., 2022) as well as other variants such as Decomposed Prompting (Khot et al., 2023) and Least-to-Most (Zhou et al., 2023) guide LLMs to decompose a complex question into a sequence of thoughts and then synthesize an answer by resolving them methodically. It has been shown that Self-Consistency (CoT-SC) (Wang et al., 2022) can be used to augment such methods by generating multiple thought sequences and then selecting the most accurate answer through majority voting. Recent meta-prompting techniques (Suzgun & Kalai, 2024) employ a uniform, task-independent prompting framework across multiple tasks, enabling a single LLM to iteratively refine its responses and dynamically adapt to diverse input queries. The Buffer of Thoughts (BoT) (Yang et al., 2024) framework extracts task-specific information, uses it to retrieve relevant thought templates from its meta-buffer, and then instantiates them with more task-specific reasoning structures before continuing with the reasoning process.

**Refinement.** Closed-loop approaches that allow an LLM to interact with an external environment can help in choosing and potentially revising an action. Notable examples are ReAct (Yao et al., 2023), REFINER (Paul et al., 2023) and Self-Refine (Madaan et al., 2023). Reflexion (Shinn et al., 2023) provides further linguistic feedback based on previous attempts while AdaPlanner (Sun et al., 2023) also incorporates positive and negative feedback of an individual trajectory. Reason for future, act for now (RAFA) (Liu et al., 2024) develops further by planning a trajectory, gathering feedback for the potential planned actions, and then revising the trajectory based on the feedback.

**Tree search.** Thoughts are individual ideas or steps in reasoning, and when connected together, they can be modeled as a tree data structure. Tree search algorithms can then be used to explore the tree of thoughts and optimize the search for a final answer. In "Tree of Thoughts" (ToT), the authors utilize a value function that compares different branches to describe both DFS and BFS flavors of a guided tree-search (Yao et al., 2024). The closely related "Graph of Thoughts" (GoT) approach relaxes the assumption of a strict tree structure (Besta et al., 2024). Reasoning via Planning (RAP) (Hao et al., 2023) augments LLMs with a world model and employs Monte Carlo Tree Search (MCTS)-based planning to reduce the search complexity. Language Agent Tree Search (LATS) (Zhou et al., 2024) extends this concept by leveraging environment interactions, thereby eliminating the need for a world model. ReST-MCTS* (Zhang et al., 2024) builds on this line by integrating process-level rewards into MCTS, by identifying high-quality reasoning traces.

**Particle filtering.** Genetic particle filters have been used successfully across a large variety of domains, ranging from vehicle routing (Marinakis & Marinaki, 2010), to fuzzy cognitive maps in time series forecasting (Salmeron & Froelich, 2016), to the positioning of industrial robots (Li et al., 2022). Near-universally, the efficacy of a genetic particle optimization approach has, in these contexts, been demonstrated on standard artificial neural networks, however, to the best of our knowledge, it has not been employed on LLMs.

**Key differences.** FOA exhibits fundamental differences from all aforementioned methods. First, it does not require extensive or intricate prompt engineering, unlike AoT (Sel et al., 2024) and meta prompting (Suzgun & Kalai, 2024). Additionally, it is well-suited for sequential decision-making tasks that require interactions with an environment, such as web navigation (Yao et al., 2022; Arora et al., 2022; West et al., 2009), which can be challenging for methods such as CoT (Wei et al., 2022) and BoT (Yang et al., 2024). Furthermore, FOA distinguishes itself from approaches like Reflexion (Shinn et al., 2023) and RAFA (Liu et al., 2024) by utilizing a refinement strategy based on a genetic-type particle filter, instead of linguistic feedback. Lastly, unlike tree-search-based methods (Yao et al., 2024; Zhou et al., 2024; Besta et al., 2024), FOA employs a more principled approach to search tree exploration, achieving a better balance between exploration and exploitation. This structured approach also grants FOA precise control over the tree's width and depth, leading to predictable latency and cost. Please refer to Fig 8 for a side-by-side comparison of the branching mechanism of ToT and FLEET OF AGENTS.

## 3. FLEET OF AGENTS

### 3.1. Overview of FOA

Fig. 3 provides a pictorial overview of FOA. FOA spawns a fleet of $n$ agents that collectively search for a solution. The genetic filtering search has a *mutation phase* during which each agent explores the search space autonomously. Specifically, it tracks the $n$ agent states and allows $k$ independent mutation steps before evaluating the values of the current states. During the *selection phase*, we resample, with replacement, the population of agents. The *resampling* mechanism is based on a heuristic value function and allows us to optimize the trade-off between exploration and exploitation. The FOA algorithm is comprehensively described in Algorithm 1 in the Appendix.

**Preliminaries.** The fleet consists of $n$ agents. At time $t$ agent $i$ has the state $s_{i,t}$. When choosing an action, the agent samples from its policy $a_{i,t} \sim \pi_a(a|s_{i,t})$. The agent then transitions to a new state, following the dynamics of the environment $s_{i,t+1} \sim \mathbb{P}[s|a_{i,t}, s_{i,t}]$. With each agent

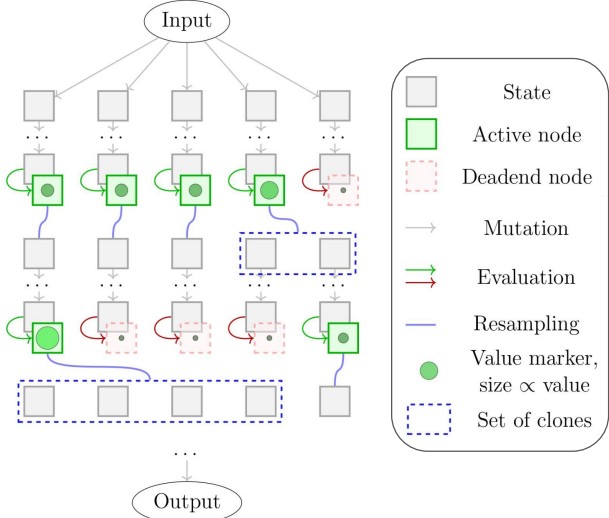

Figure 3: Fleet-of-Agents (FoA) comprising $n = 5$ agents that think autonomously for $k$ steps and are then resampled to focus the search on promising regions.

searching for the solution, the fleet jointly discovers more and more of the state space. We describe the current set of states at timestep $t$ as $S_t = \{s_{i,t}\}, i = 1..n$ and the set of all states visited so far as $\hat{S}_t = \bigcup_{\hat{t}} S_{\hat{t}}, \hat{t} = 1..t$.

We assume that we can identify solutions when they are found, i.e., we can decide whether a state $s$ is a solution, $\mathbb{1}_{solution}(s)$. Depending on the task, we may also be able to identify invalid states, failed tasks, and other forms of dead-ends; we denote them as terminal states $\mathbb{1}_{terminal}(s)$.

For some tasks, we can stop as soon as a solution is found; for other tasks, we may instead collect a set of solution candidates and stop the search only if we run out of time or sampling budget. We assume we are given access to a value function $v(s)$ that can guide the search. For a solution state $s$, the value function represents the utility of the solution. For other states, $s$, the value function is a heuristic considering the uncertainty of eventually reaching a solution and its expected utility. Accordingly, the value of terminal, non-solution states is 0.

### 3.2. Genetic particle filtering

Each agent in the fleet of agents acts autonomously and tries to choose the best action given its current state. After $k$ independent steps, we use the value function to resample the set of states and allocate more agents to high-value states. Our algorithm implements a genetic-type particle filter that captures the dynamics of a population of particles, i.e., agents, with a series of mutation and selection steps.

**Mutation phase.** During the mutation phase, each agent in FoA independently samples state transitions. To simplify the notation, we introduce a model $\pi$ which captures both

the stochasticity of the decision of the agent as well as the response of the environment, $s_{i,t+1} \sim \pi(s|s_{i,t})$ We use the same notation for multi-step state transitions by marginalizing over intermediate states, i.e., $s_{i,t+k} \sim \pi(s_{i,t+k}|s_{i,t})$. After each mutation step, we can check whether a solution has been found and decide to stop the search. Following the concept of genetic filtering, we apply two optimizations:

• **Enforced Mutation**: The agent must mutate its state; it can not remain stationary.

• **Sudden Death**: Depending on the task, some solution attempts may result in invalid states. For example, in a crossword puzzle, an agent might propose a word with an incorrect number of characters, resulting in an ill-defined state. It is standard practice, across algorithms (e.g. ToT, GoT), to prevent the search from continuing in such cases. For FoA, we use a simple filtering mechanism: when an agent enters an invalid state, it is deleted, and another agent is randomly duplicated to fill the gap. The replacement agent is chosen uniformly at random, avoiding additional calls to the value function.

**Selection phase.** The selection phase resamples the population of agents based on an importance sampling mechanism. We observe the value estimates $v(s_{i,t})$ for all current states and calculate a resampling weight $p_{i,t}$. This framework can capture many resampling schemes, such as linear, greedy, or exponential weighting of values:

$$p_{lin}(s_{i,t}) = \alpha v(s_{i,t}) + \beta,$$

$$p_{exp}(s_{i,t}) = \exp\left(\frac{v(s_{i,t})}{\beta}\right),$$

$$p_{greedy}(s_{i,t}) = \begin{cases} 1, & \text{if } s_{i,t} = \arg\max_{s_{j,t},j=1..N} v(s_{j,t}) \\ 0, & \text{otherwise} \end{cases}.$$

We then resample, with replacement, to select a new set of agent states:

$$p_t(s) = \sum_{i=1}^{N} \frac{p_{i,t}}{\sum_{j=1}^{N} N p_{j,t}} \delta_{s_{i,t}}(s), \text{categorical resampling}$$

$$\hat{s}_{i,t} \overset{iid}{\sim} p_t(s), \text{resampling with replacement}$$

$$s_{i,t} = \hat{s}_{i,t}, i = 1..N$$

**Backtracking.** Additionally, by keeping track of a history of states and the associated value function estimates, we extend the resampling process with an intuitive backtracking mechanism. Backtracking undoes localized mistakes and allows our fleet of agents to recover from a catastrophic scenario in which all agents might have made a wrong decision. Instead of resampling states from the set of current states $S_t$, we consider all previously visited states $\hat{S}_t$. To incentivize the fleet of agents to push forward and explore new regions

of the state space, we introduce a discount factor $\gamma$. The value of a state that was visited $t$ timesteps ago is discounted by $\gamma^t$. The mutation phase of our genetic particle filter comprises $k$ steps of individual exploration, before evaluating and resampling in the selection phase. Therefore, we may not have a value estimate for all previously visited states $\hat{S}_t$. Depending on the task, and the corresponding cost of computing the value $v(s)$, we may limit the backtracking mechanism to a subset of $\hat{S}_t$ which only contains states with a known value estimate.

# 4. Experiments

We assess the effectiveness of our proposed FoA framework through comparisons with representative SOTA methods on a judicious mix of tasks that require a variety of reasoning, planning, and general problem-solving skills. Additional experimental details, e.g., hyperparameter tuning, additional results, etc. are present in Appx. C. The resources for reproducing our experiments are available at https://github.com/au-clan/FoA.

**Base model.** Following convention in the literature, we use GPT-4[1] as the base model for the main results presented in this paper. To showcase the *generalizability* of our findings, we report results with other base models, namely, GPT-3.5, Llama3.2-11B, and Llama3.2-90B, in Appx. D. We also use these models for the model and ablation analyses (§ 5 and § 6), primarily for practical cost-related reasons.

**Number of runs.** For cost reasons, experiments with GPT-4 were run only once. However, for other base models (results in Appx. D), we run each experiment 5 times and report the mean of the evaluation metrics.

**Prompts.** Unlike all existing works, we *do not craft custom prompts* for FoA, but instead, we reuse the prompts (cf. Appx. C.3 for details) provided by ToT (Yao et al., 2024) for the "Game of 24" and "Mini Crosswords" tasks, LATS (Zhou et al., 2024) for the "WebShop" task, and ReST-MCTS* (Zhang et al., 2024) for the "SciBench" task. This ensures a *direct and fair comparison* of the reasoning abilities of the benchmarked frameworks and controls for the impact of prompt quality, which is a confounder.

**Baselines.** We only compare with methods that have made their code available for the tasks benchmarked in this study (cf. Appx. C.2 for details). Thus, we do not compare with LLMCompiler (Kim et al., 2024), TouT (Mo & Xin, 2024), RAP (Hao et al., 2023), and RecMind (Wang et al., 2024b). Moreover, we exclude BoT (Yang et al., 2024), where al-

---

[1]Owing to the exorbitant cost of running GPT-4 (details in App. C.3), we use GPT-3.5 instead for WebShop (Yao et al., 2022) and SciBench (Wang et al., 2024a).

Table 1: Comparing FoA with previous methods using *success rate* (↑ better) and *cost* (↓ better) on the Game of 24 task (base model: GPT-4). The best performance is shown in `blue` whereas the second best is shown in `orange`. Owing to its exorbitant cost ($\simeq$ 600 US$), we could not run RAFA (Liu et al., 2024) (not shown in the table).

| Method | Success Rate (%) | Cost (US$) |
|---|---|---|
| IO | 6.0 | **0.65** |
| CoT (Wei et al., 2022) | 6.0 | **6.98** |
| CoT-SC (Wang et al., 2023) | 10.0 | 49.40 |
| AoT (Sel et al., 2024) | 49.0 | 20.98 |
| ToT (Yao et al., 2024) | **74.0** | 75.02 |
| GoT (Besta et al., 2024) | 63.0 | 70.01 |
| ReST-MCTS* (Zhang et al., 2024) | 38.0 | 104.62 |
| FoA **(Present Work)** | **76.0** | 62.93 |

though the code is available, an important resource (the meta-buffer) to reproduce their results is unavailable. We include ReST-MCTS* (Zhang et al., 2024) as its code is available for SciBench. However, since no implementation was provided for other tasks, we re-implemented the method ourselves for Game of 24 to enable a broader comparison.

## 4.1. Game of 24

**Task and data.** Game of 24 is a mathematical puzzle, where four numbers are given, and the objective is to form an arithmetic expression that equals 24 using each number exactly once. The benchmark data consists of 1362 puzzles. Following ToT (Yao et al., 2024), we use the puzzles indexed 901-1000 as the test set (cf. Appx. C.1 for details).

**Evaluation metrics.** We use *success rate*, i.e., the percentage of solved puzzles, to evaluate the quality of the benchmarked methods. For efficiency, we use cost (in US$).

**Baselines.** We compare FoA with: (1) Standard IO prompting, (2) CoT (Wei et al., 2022), (3) CoT-SC (Wang et al., 2023), (4) AoT (Sel et al., 2024), (5) ToT (Yao et al., 2024), (6) GoT (Besta et al., 2024), and (7) RAFA (Liu et al., 2024). Owing to the unavailability of their code for Game of 24, we do not compare with LATS (Zhou et al., 2024).

**Results.** Table 1 shows that FoA outperforms all existing baselines and achieves the best quality. Taking GPT-4 as a baseline, FoA achieves a whopping 70% improvement in quality. On the one hand, IO and CoT (Wei et al., 2022) are the most cost-effective methods, their success rate is extremely low at just 6%. On the other hand, sophisticated reasoning frameworks like ToT (Yao et al., 2024) achieve a high success rate (74%) but also incur a high cost (75$). Striking a good balance between exploration vs. exploitation, our FoA obtains a *2% absolute improvement in quality over the second best method, ToT, simultaneously lowering the cost requirement by 25%.*

Table 2: Comparing FOA with previous methods using *overlap* (↑ better) and *cost* (↓ better) on the Crosswords task (base model: GPT-4). The best performance is shown in blue whereas the second best is shown in orange.

| Method | Overlap (%) | Cost (US$) |
|---|---|---|
| IO | 36.8 | **0.51** |
| CoT (Wei et al., 2022) | 39.4 | **1.06** |
| CoT-SC (Wang et al., 2023) | 39.4 | 2.82 |
| ToT (Yao et al., 2024) | 39.7 | 48.99 |
| GoT (Besta et al., 2024) | **41.2** | 30.28 |
| FoA (**Present Work**) | **46.0** | 12.94 |

### 4.2. Mini Crosswords

**Task and data.** Mini Crosswords is a puzzle, where, given 5 vertical and 5 horizontal clues, the objective is to use the clues to identify answers and place them on a $5 \times 5$ crossword board. The benchmark data consists of 156 puzzles. Following ToT (Yao et al., 2024), we use the puzzles 0, 5, . . . , 90, and 95 as the test set (cf. Appx. C.1 for details).

**Evaluation metrics.** For quality, we use *overlap*, i.e., the percentage of correct letters in the proposed solution. For efficiency, we compute the cost (in US$).

**Baselines.** We compare FOA with: (1) Standard IO prompting, (2) CoT (Wei et al., 2022), (3) CoT-SC (Wang et al., 2023), (4) ToT (Yao et al., 2024), and (5) GoT (Besta et al., 2024). Owing to the unavailability of their code for Mini Crosswords, we do not compare with AoT (Sel et al., 2024).

**Results.** Table 2 shows that FOA outperforms all existing baselines and achieves the best quality. Once again, IO and CoT (Wei et al., 2022) are the most cost-effective methods. Moreover, they obtain good performance with their quality being comparable to ToT (Yao et al., 2024) and GoT (Besta et al., 2024) at just a fraction (2.5%) of the cost. Notably, our FOA reports the best cost-quality trade-off among all the benchmarked methods. We obtain a *5% absolute improvement in quality over the second best method, GoT, simultaneously lowering its cost requirement by 60%.*

### 4.3. WebShop

**Task and data.** WebShop (Yao et al., 2022) is a simulated e-commerce website environment, where, given a textual instruction specifying a product and its properties, the objective is to find the product by navigating webpages using a variety of actions and purchase it. The benchmark data consists of 12,087 subtasks. Following (Yao et al., 2023; Zhou et al., 2024; Shinn et al., 2023), we use 50 randomly sampled subtasks as the test set (cf. Appx. C.1 for details).

Table 3: Comparing FOA with previous methods using *average score* (↑ better) and *cost* (↓ better) on the WebShop task (base model: GPT-3.5). The best performance is shown in blue whereas the second best is shown in orange.

| Type | Method | Avg Score | Cost (US$) |
|---|---|---|---|
| Super-
vised | IL (Yao et al., 2022) | 59.9 | NA |
| | IL+RL (Yao et al., 2022) | 62.4 | NA |
| | WebN-T5 (Gur et al., 2023) | **61.0** | NA |
| | WebGUM (Furuta et al., 2024) | **67.5** | NA |
| In-
context | Act (Yao et al., 2023) | 58.1 | **0.10** |
| | ReAct (Yao et al., 2023) | 48.7 | **0.17** |
| | Reflexion (Shinn et al., 2023) | 56.3 | 0.65 |
| | LASER (Ma et al., 2023) | 57.2 | 0.41 |
| | LATS (Zhou et al., 2024) | **66.1** | 232.27 |
| | FoA (**Present Work**) | **75.6** | 1.68 |
| | Hum. experts (Yao et al., 2022) | 82.1 | NA |

**Evaluation metrics.** The quality of a purchase is assessed using an environment-generated reward, which measures the percent overlap between the purchased product and the user-specified attributes. We use *average score*, i.e., the average of subtask rewards, to evaluate the quality of the benchmarked methods. For efficiency, we use cost (in US$).

**Baselines.** We compare FOA with: (1) Act (Yao et al., 2023), (2) ReAct (Yao et al., 2023), (3) Reflexion (Shinn et al., 2023), (4) LASER (Ma et al., 2023), (5) LATS (Zhou et al., 2024), and (6) multiple fine-tuned models from Yao et al. (Yao et al., 2022). We also use the performance of human experts as an upper bound for quality. Owing to the unavailability of their code for WebShop, we do not compare with RAP (Kagaya et al., 2024).

**Results.** Table 3 shows that FOA outperforms all existing baselines, even the supervised fine-tuned models (Yao et al., 2022; Gur et al., 2023; Furuta et al., 2024), and achieves the best quality. While Act (Yao et al., 2023) is by far the cheapest method (0.1$), it obtains a moderate average score (58.1%). LATS (Zhou et al., 2024), on the other hand, obtains a better average score (66.1%), it suffers from an exorbitant cost footprint (232$). Yet again, our FOA achieves the best cost-quality trade-off: obtaining a *10% absolute improvement in quality over the second-best method, LATS, requiring only 1% of its cost.*

### 4.4. SciBench

**Task and data.** SciBench (Wang et al., 2024a) is a benchmark suite of scientific reasoning tasks, where the objective is to answer complex, multi-step scientific questions that require integrating knowledge, reasoning over evidence, and applying scientific principles. The benchmark data consists of questions across physics, chemistry, and biology domains,

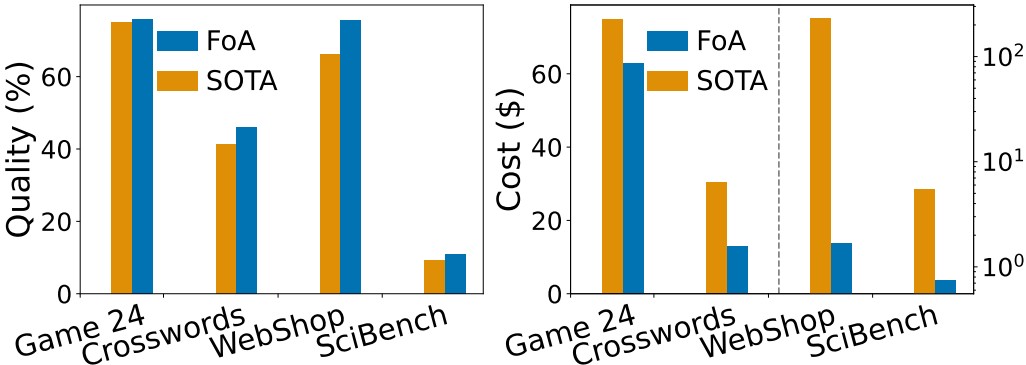

Figure 4: Comparing (Left) quality and (Right) cost of FOA with the second most efficacious method (labeled SOTA in the plot) on each benchmark task. Note that the cost plot (Right) uses a split y-axis (indicated by the vertical dashed line) to accommodate the wide range of cost values: a linear scale for Game 24 and Crosswords, and a logarithmic scale for WebShop and SciBench.

with each domain containing a different number of questions. We use the first 15% of each domain's questions as the validation set and the remaining 85% as the test set.

**Evaluation metrics.** The quality is measured using *accuracy*, which reflects the proportion of final responses matching exactly with the oracle answer. For efficiency, we use cost (in US$).

**Baselines.** We compare FOA with: (1) Standard IO prompting, (2) CoT (Wei et al., 2022), (3) CoT-SC (Wang et al., 2023), (4) ToT (Yao et al., 2024) and (5) ReST-MCTS* (Zhang et al., 2024).

**Results.** In Table 4, we present the averaged results across domains (cf. Table 7 for domain-specific results). FOA outperforms all existing baselines and achieves the best quality. Even though, IO is by far the cheapest baseline (0.01$) its accuracy (5.9%) is almost half of FOA. Finally, FOA achieves the best cost-quality trade-off: obtaining a *1.7% absolute improvement in quality over the second-best method, ReST-MCTS*, requiring only 14% of its cost.*

Table 4: Comparing FOA with previous methods using *accuracy* (↑ better) and *cost* (↓ better) on the SciBench task (base model: GPT-3.5). The best performance is shown in blue whereas the second best is shown in orange .

| Method | Accuracy (%) | Cost (US$) |
|---|---|---|
| IO | 5.9 | **0.01** |
| CoT (Wei et al., 2022) | 8.4 | **0.03** |
| CoT-SC (Wang et al., 2023) | 8.7 | 0.64 |
| ToT (Yao et al., 2024) | 7.2 | 10.34 |
| ReST-MCTS* (Zhang et al., 2024) | **9.3** | 5.11 |
| FoA (**Present Work**) | **11.0** | 0.80 |

## 5. Model Analysis

**FOA vs. SOTA.** To better understand the improvements obtained by FOA, we compare it with the second most efficacious method (SOTA hereafter) on all four benchmark tasks. Specifically, ToT (Yao et al., 2024), GoT (Besta et al., 2024), LATS (Zhou et al., 2024) and ReST-MCTS* (Zhang et al., 2024) are the SOTA methods for the Game of 24, Crosswords, WebShop, and SciBench tasks, respectively. As stated in § 4, we use GPT-4 as the base model for Game of 24 and Crosswords, and GPT-3.5 for WebShop and SciBench. Fig. 4 shows that FOA not only achieves the best quality but also the lowest cost on all tasks. **On average, FOA obtains a quality improvement of $\simeq 5\%$ while requiring only $\simeq 35\%$ of the cost of SOTA methods.**

**Trade-off between Cost and Quality.** We analyze the trade-off between cost and quality for the top five methods, namely, ToT (Yao et al., 2024), GoT (Besta et al., 2024), RAFA (Liu et al., 2024), ReST-MCTS* (Zhang et al., 2024) and FOA, in the Game of 24 task. For cost reasons, we use GPT-3.5 as the base model. Following RAFA (Liu et al., 2024), we allow each method to perform multiple trials ($\Delta$) and consider them successful if they generate a valid solution in any of the $\Delta$ trials. To ensure fairness in the allocated resources, we allocate a fixed budget of 5$ per method, which governs the number $\Delta$ of trials for each method (5 for ToT and GoT, 6 for FOA, and 10 for RAFA).

Fig. 5 (right) shows that FOA substantially outperforms the existing SOTA methods at all possible price points. Based on the portrayed trends, ToT might be able to close the quality gap or even surpass FOA with a higher budget, however, FOA is always favorable for resource-constrained settings. **Overall, FOA achieves the best cost-quality trade-off.**

**Trade-off between Model-size and Quality.** Fig. 5 (left) shows that on their own, both the 11B and 90B Llama3.2

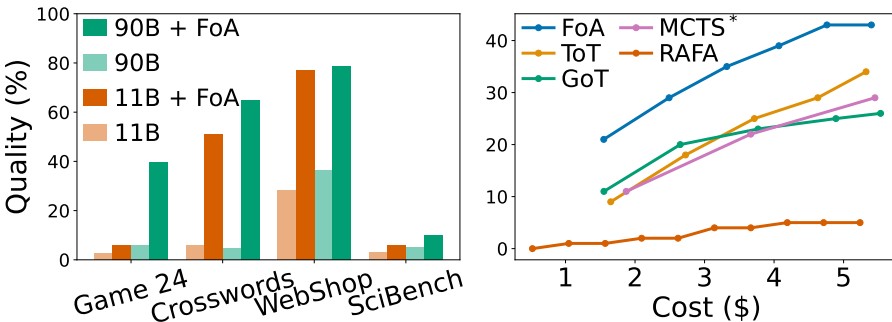

Figure 5: Evaluating the trade-off between (Left) model size and quality on the benchmarked tasks with Llama3.2-11B and 90B as base models, and (Right) cost and quality of representative SOTA methods with GPT-3.5 on Game of 24.

models (Grattafiori et al., 2024) achieve poor quality on the benchmarked tasks. However, FOA boosts the performance of both models by portraying significant (5x–6x) quality improvements. What's more, Llama3.2-11B + FOA surpasses the larger Llama3.2-90B model. Overall, **FOA enables smaller models to obtain comparable or even better performance than larger models**, thereby bridging the gap between their reasoning abilities.

## 6. Ablation Analysis

In this section, we report results on Game of 24 with GPT-3.5 and Llama3.2 (11B&90B) as base models. We observed similar trends for the other three tasks, results in Appx. D.2.

**Impact of the Selection phase.** We remove the selection phase by setting the resampling frequency $k = 0$. Thus, each agent constantly remains in its own mutation phase, and independently works towards its goal until a solution is found, a terminal state is found, or time runs out. As expected, FOA (without selection) obtains an extremely low success rate but is also cheaper (Fig. 6a). This is because, without the selection phase, the fleet does not evolve into a composition of more high-value states with each time step.

**Impact of Resampling.** Instead of using a resampling strategy, we assign each agent to the highest-value state during the selection phase. If there are multiple such states, the first one is randomly chosen by all agents. When there are multiple promising states, a meaningful resampling strategy may decide to explore all of them in the next time step, however, retaining only the highest-value state reduces the fleet diversity. Thus, FOA (no/max resampling) obtains a slightly lower success rate but also incurs a lower cost (Fig. 6b).

**Impact of the Backtracking mechanism.** We vary the discount factor $\gamma$ to study two aspects of the backtracking mechanism: (1) $\gamma = 0$ (no backtracking) and (2) $\gamma = 1$ (no decay). When $\gamma = 0$, resampling from a past state is not permitted, and thus, FOA cannot undo localized mistakes, resulting in a lower success rate but also lower cost (Fig. 6c).

With $\gamma = 1$, all past states are considered during resampling, thereby increasing the spectrum of states to explore but at the risk of overwhelming the resampling mechanism with noisy value estimates of (relatively older) past states. This explains the reduction in success rate and a slight increase in cost owing to potentially futile explorations (Fig. 6c).

**Impact of Caching.** Similar to ToT (Yao et al., 2024) and LATS (Zhou et al., 2024), FOA utilizes a caching mechanism (details in Appx. C.3) to enhance its cost efficiency, which ensures that a given state is evaluated only once by an LLM. As expected, disabling the cache leads FOA to incur a higher cost, but interestingly, it also leads to a lower success rate (Fig. 6d). We hypothesize that frequent re-evaluations of the same state while solving a puzzle might introduce inconsistencies, further disrupting the flow of reasoning structures, thereby leading to a reduction in quality.

**Impact of Batching.** Similar to ToT (Yao et al., 2024) and LATS (Zhou et al., 2024), FOA utilizes a batching mechanism (details in Appx. C.3) to enhance its cost efficiency, which groups many individual LLM calls into a single combined call (one input prompt leading to many output responses). Disabling batching leads FOA to incur a higher cost with no noticeable impact on quality (Fig. 6e).

In sum, Fig. 6 shows that each component of the FOA framework has an overall positive impact on its performance.

## 7. Discussion and Concluding Insights

### 7.1. Summary of Findings

**FOA achieves better quality than all existing methods.** Based on the results presented in § 4 and Appx. D, FOA consistently outperforms all existing SOTA methods across all benchmarked tasks and base models.

**FOA achieves a better cost-quality trade-off.** Our results show that FOA is cost-efficient, much more so than other SOTA reasoning frameworks. Moreover, our analysis in

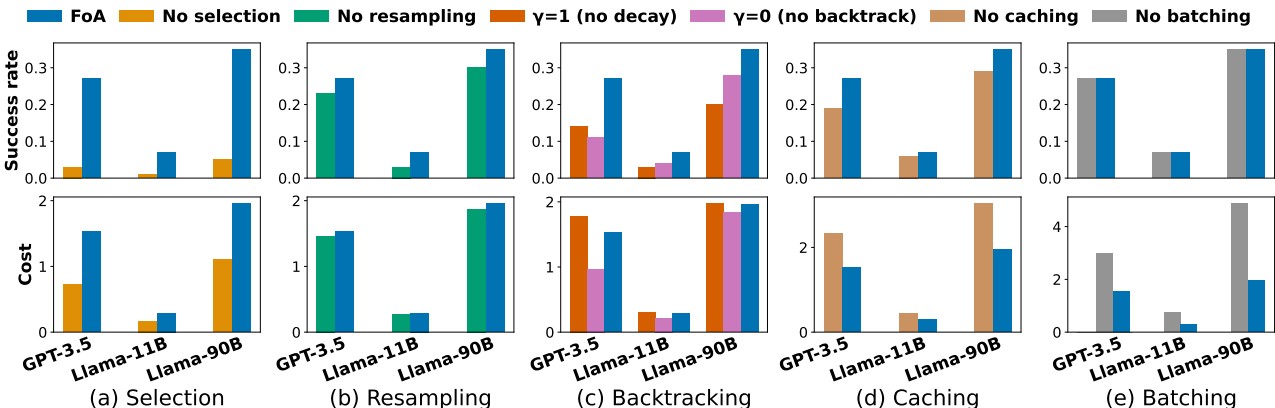

Figure 6: Ablation analysis to study the impact of (a) Selection phase, (b) Resampling, (c) Backtracking, (d) Caching, and (e) Batching on the performance of FOA using the Game of 24 task with GPT-3.5, Llama3.2-11B, and 90B as base models.

§ 5 reveals that FOA substantially outperforms all SOTA methods at all possible price points.

**Other advantages.** Beyond better performance, FOA offers many important practical advantages. First and most importantly, FoA is not a prompting scheme but a runtime. Unlike existing prompt-based frameworks, e.g. ToT (Yao et al., 2024), GoT (Besta et al., 2024), etc., FOA does not require custom-crafted prompts, but, instead, can be used in combination with any existing agent or prompting strategy. Next, FOA offers precise control over the tree's width ($n$ agents) and depth ($t$ steps), leading to predictable latency and cost. In particular, it is possible to tune the size $n$ of the fleet to the available resources, such as an optimal batch size for a given hardware configuration. Finally, as evidenced by a smaller standard error (Appx. D), FOA generates consistent responses across multiple runs and is relatively more stable than other methods.

### 7.2. Limitations and Future work

**Constant fleet size.** Currently, we assign a fixed number $n$ of agents to each task. However, it may be advantageous to allocate more agents to more difficult tasks to enhance sample efficiency. In the future, we would like to explore the possibility of an adaptive fleet size.

**Resampling mechanisms.** One avenue for improvement is the design of more intelligent value functions, for example, pooling information from neighboring states and smoothing predictions. Building on precise value estimates, we can investigate different resampling mechanisms, to tune FoA towards either more risk-seeking or careful behavior, i.e., different tradeoffs between exploration and exploitation.

**Fleet organization.** Currently, we consider a homogenous fleet of identical agents. In the future, we would like to introduce further coordination between the individual agents with a hierarchical organizational structure. For instance, what if agents could spawn other agents in a nested particle filtering framework? Finally, as a runtime that embraces modular compatibility with any agent, FoA will also enable research into more complex fleet compositions.

Overall, we hope that our work will motivate further research in the combination of (genetic) filtering algorithms for the orchestration of AI agents.

### Acknowledgements

We thank Anja Šurina, Bhavyajeet Singh, André Charneca, Laurent Bindschaedler, Krishna Gummadi, Chris Schwiegelshohn, and Niket Tandon for insightful discussions. West's lab is partly supported by grants from the Swiss National Science Foundation (200021_185043 and 211379). Arora's lab is partly supported by grants from the Novo Nordisk Foundation (NNF24OC0099109), the Pioneer Centre for AI, and EU Horizon 2020 (101168951). We also gratefully acknowledge generous gifts from Google and Microsoft.

### Impact Statement

Throughout an LLM's lifecycle, the majority of costs are incurred during inference rather than training (Fu et al., 2024). Thus, it is crucial to benchmark the cost incurred by various SOTA reasoning frameworks. Our work is the first to report, analyze, and include a discussion on the trade-off between cost and quality of a variety of SOTA LLM reasoning frameworks. To the best of our knowledge, it is also the first to propose the use of genetic-style algorithms in the context of agentic AI. We hope that this work will spark further discussions on the cost-efficiency of LLM reasoning frameworks and eventually lead to the development of practically feasible and more sustainable AI technologies.

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

# A. Additional Related Work

AI agents extend the capabilities of LLMs by integrating external tools or orchestrating collaborations between multiple LLMs. To solve a task, such an AI agent may take many individual steps, usually one depending on the other, e.g., querying a database, searching with a web browser, running computations in a code interpreter, etc. There exists a diverse and quite fragmented ecosystem of frameworks and libraries that motivate specific interaction patterns or are designed to offer the flexibility to implement new forms of collaboration. A well-established library is LangChain (Chase, 2022), but many practitioners chose to accompany their research with a custom solution. Cameleon (Lu et al., 2023), Camel (Li et al., 2023), HuggingGPT (Shen et al., 2023), AutoGPT (Richards, 2023), BabyAGI (Nakajima, 2023), MetaGPT (Hong et al., 2023), Flows (Josifoski et al., 2023), and AutoGen (Wu et al., 2023), all these works present some form of blueprint or framework for building AI agents.

Building on these works, creating highly sophisticated AI agents is possible. Nevertheless, even if an action is only taken after many steps of deliberation and careful consideration, the agent must ultimately follow a sequential chain of actions on its way toward a solution. This is fundamentally similar to structured reasoning, i.e., the agent approaches a solution in smaller discrete steps, each step following logically from the previous steps. In many scenarios, multiple actions may be promising; the agent is faced with uncertainty and a large state space to explore. In this setting, the perspective of a single agent is necessarily a myopic and localized worldview. We overcome this limitation by instantiating a fleet of individual agents and coordinating a joint search process between them. This research is orthogonal to existing work on optimizing AI agents: Instead of a novel way to build or improve a specific AI agent, we implement a runtime that can readily accommodate any existing agent and multiply its capabilities.

# B. FLEET OF AGENTS: Additional Details

---

**Algorithm 1** Fleet of Agents: A genetic particle filter.

---

1: Initialize
2: $t \leftarrow 0$
3: $s_{i,t} \leftarrow \mathbf{x}_0, i = 1..N$ {setting initial state}
4: **loop**
5:     **Mutation phase**
6:     $s_{i,t+k} \leftarrow$ Call Algorithm 2 with $s_{i,t}$, $i = 1..N$ and $S_t$
7:
8:     **Selection phase**
9:     $s_{i,t+k} \leftarrow$ Call Algorithm 3 with $S_{t+k}$
10:
11:     $t \leftarrow t + k$
12: **end loop**

---

## B.1. Comparison of FLEET OF AGENTS with a standard tree-search algorithm

We posit that any multi-step structured reasoning prompt will rely on two flavors of prompts: *mutation steps*, in which the current state of the reasoning process is advanced, and *evaluation steps*, in which different states are judged and compared. The key difference between structured reasoning algorithms lies in how these steps are orchestrated. In Fig. 8, we show a standard tree search algorithm, based on one of the Tree-of-Thoughts (ToT) algorithms, side by side with FLEET OF AGENTS.

At every step during the search for a solution, ToT keeps track of $b$ thoughts, generates $c$ follow-up thoughts for each, estimates their value, and from the $cb$ candidate thoughts, selects the $b$ best. It is important to note:

● Mutation and evaluation prompts are called in lockstep; each new thought is immediately evaluated.

● At every step, $cb - b$ candidate thoughts are discarded. Depending on the parametrization, the majority of mutation and value calls correspond to thoughts that immediately become dead ends.

● The number of promising states selected and the number of follow-up thoughts are fixed, resulting in a constant branching factor.

---

**Algorithm 2** Mutation Phase

---

1: **Input:** List of current states $s_{i,t}$, $i = 1..N$ and states $S_t$ at time $t$
2: **Output:** List of states $s_{i,t+k}$, $i = 1..N$ at time $t + k$
3: **for** $j = 1$ to $k$ **do**
4:     Each agent independently mutates its state
5:     $s_{i,t+1} \sim \pi(s|s_{i,t}), i = 1..N$
6:     $t \leftarrow t + 1$
7:     Check for solution
8:     **if** $\exists s \in S_t : \mathbb{1}_{solution}(s)$ **then**
9:         We have found a solution: early exit
10:     **end if**
11:     Identify and prune terminal states,
12:     resample across all viable states $S_t$ discovered so far
13:     $B \leftarrow \{s_{i,t}|\mathbb{1}_{terminal}(s_{i,t})\}$
14:     $G \leftarrow \{s|s \in S, s \notin B\}$
15:     **for** $i \in B$ **do**
16:         $\hat{s}_i \sim \text{Uniform}(G)$
17:         $s_{i,t} \leftarrow \hat{s}_i$
18:     **end for**
19: **end for**

---

---

**Algorithm 3** Selection Phase

---

1: **Input:** Set of states $S_t$ at time $t$
2: **Output:** Resampled list of states $s_{i,t}$, $i = 1..N$ at time $t$
3: Evaluate heuristic value function
4: $v_s \leftarrow v(s), s \in S_t$
5: Calculate the resampling distribution
6: $p_s = \exp(1\beta v_s), s \in S_t$, {resampling weight}
7: $p(s) = \frac{1}{\sum_{\hat{s} \in S} p_{\hat{s}}} \sum_{s \in S} p_s \delta_{s_{i,t}(s)}$ {categorical resampling distribution}
8: $\hat{s}_i \overset{iid}{\sim} p_t(s), i = 1..N$ {resampling with replacement}
9: $s_{i,t} = \hat{s}_i, i = 1..N$

---

This ToT algorithm closely resembles beam search using a value function as a heuristic. By comparison, FLEET OF AGENTS follows a different design philosophy. Instead of modeling the multi-step reasoning process as a graph traversal or heuristic tree search, we take inspiration from the concept of individual AI agents exploring a complex search space on their own. We believe that for many real-world applications, it is best to trust the individual agent to make local decisions based on their best judgment. Rather than exploring $c$ branches at each step, we allow our fleet of $N$ agents to make a series of educated guesses.

We evaluate the states of individual agents only every $k$ steps. Instead of selecting the $b$ best states, we then resample with replacement. If one of the agents has found an extremely promising state, it is duplicated and the search focuses on the region that agent is exploring. This corresponds to a decision to exploit the findings of this agent. Conversely, if all agents have a similar value, none of them may be duplicated during the resampling process, keeping maximum diversity in the fleet of agents. This corresponds to a decision to explore further.

Refer to the first resampling stage in Fig. 8, which shows that one agent is duplicated (replacing an agent of very low value) while three others are kept. In practice, this means that we allocate a branching factor of 2 to the duplicated agent and a branching factor of 1 to the rest of the fleet. This is a reasonable choice: When we have yet to observe a distinct difference in thought value, then it is better to explore further before concentrating the search. In the second resampling stage, we note that one thought has a much higher value than the others. The resampling creates four copies of this thought but also retains one medium-value state. In this toy scenario, ToT always explores the dynamic search tree with a fixed branching factor of 3, whereas FLEET OF AGENTS selects branching factors dynamically, ranging from 1 to 4.

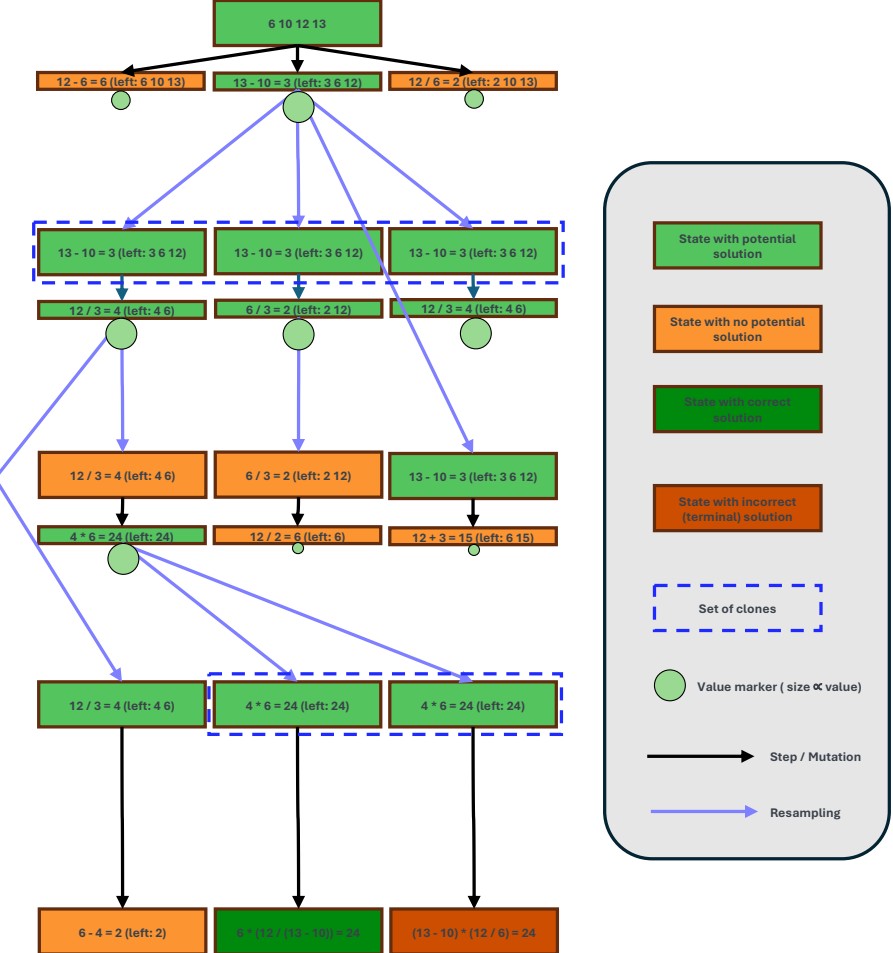

Figure 7: Example of a Fleet of Agents runtime applied to the Game of 24. Three agents are deployed, and the runtime undergoes the selection phase every $k = 1$ steps. A positive discount factor $\gamma > 0$ is retained to allow backtracking to previous states. During the resampling in the mutation phase, agents that have taken incorrect actions are corrected by replacing their state with one that has a potential solution.

Notable properties of FLEET OF AGENTS include:

• Individual decisions are made according to each agent's localized best judgment. This enables quick and efficient spread across the search space.

• Resampling (i.e., discarding some agents) occurs only every $k$ steps. Ideally, only thoughts clearly worse than their competitors are discarded.

• The resampling process provides an intuitive trade-off between exploitation and exploration. In practice, FLEET OF AGENTS can dynamically choose a branching factor between 1 (all agents are retained, maximizing exploration) and $N$ (one agent finds a highly promising state and the search is focused accordingly).

## C. Additional Experimental Details

### C.1. Detailed Task Descriptions

#### C.1.1. GAME OF 24

Game of 24 is a mathematical puzzle where the participants are presented with four numbers, and their objective is to find a combination of arithmetic operations (+-*/) to construct an arithmetic expression that uses each given number exactly once to obtain a final total of 24.

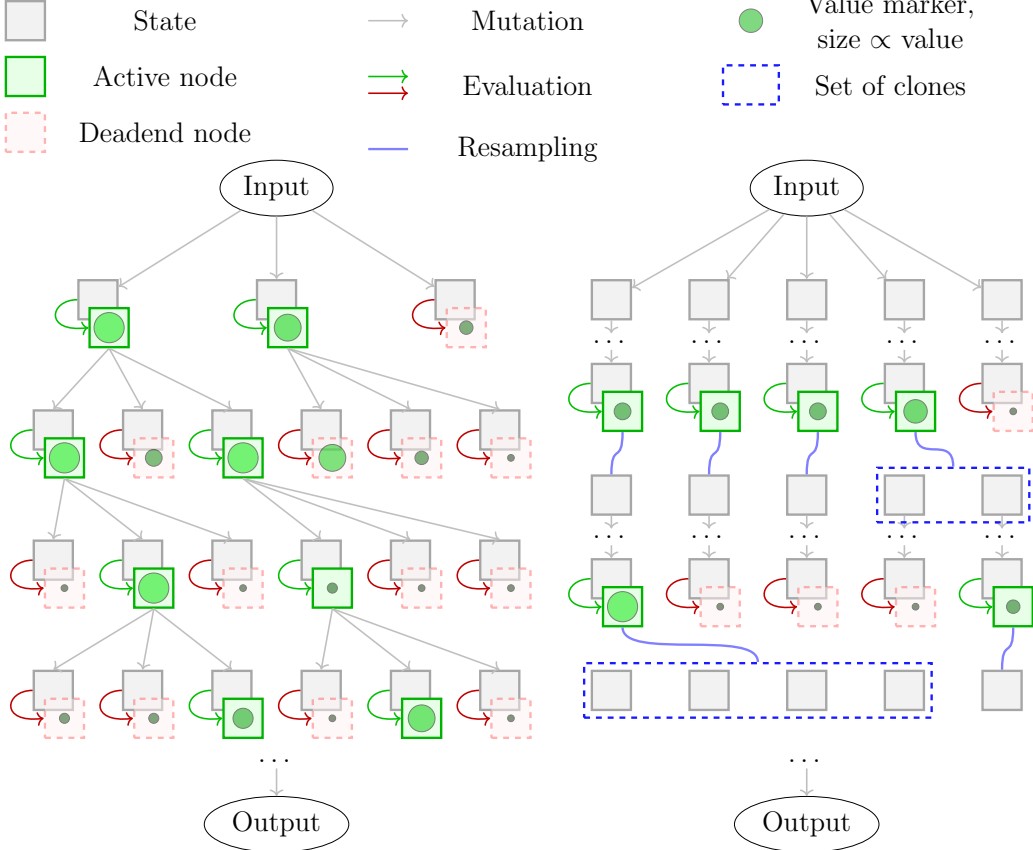

Figure 8: (Left) Search-tree for Tree-of-thoughts (ToT) that generates $c = 3$ candidates for the next thought step and maintains a set of $b = 2$ most promising states at each step. (Right) Fleet-of-Agents (FoA) comprising $N = 5$ agents that think autonomously for $k$ steps and are then resampled to focus the search on promising regions.

The benchmark consists of 1362 such puzzles scraped from 4nums.com, which are sorted in increasing order of their difficulty. The input data in each puzzle are the four initial numbers and the expected output is an equation that equals 24. Following ToT (Yao et al., 2024), we use the puzzles indexed 901–1000 as the test set. We also created a validation set from the puzzles indexed 876–900 and 1001–1025. The validation set is constructed such that, in expectation, its overall difficulty is similar to the test set.

### C.1.2. MINI CROSSWORDS

Mini Crosswords is a puzzle where participants are given 5 vertical and 5 horizontal clues. Each clue leads to a 5-letter word, and the objective is to use the clues to identify answers and place them on a $5 \times 5$ crossword board. We use the percentage of correct letters to measure the quality of a proposed crossword solution. For a letter to be correct, it has to match both the letter and its position on the ground truth board.

The benchmark constitutes 156 such puzzles scraped from GooBix. Following ToT (Yao et al., 2024), we use the puzzles 0, 5, ..., 90, and 95 as the test set. We also created a validation set from the puzzles indexed 3, 8, ..., 93, and 98.

### C.1.3. WEBSHOP

WebShop (Yao et al., 2022) is a simulated e-commerce website environment comprising 1.18 million real-world products and 12,087 crowd-sourced textual instructions. The participants are provided with textual instructions specifying a product and its properties, and their objective is to find and purchase the product by navigating webpages using a variety of actions.

The benchmark data consists of 12,087 subtasks. We noticed that the website environment randomizes the set of subtasks upon every initialization. Thus, to fix the test set across all the experiments and methods benchmarked in this study, we add a fixed random seed to the website environment. Following (Yao et al., 2023; Zhou et al., 2024; Shinn et al., 2023), we

use 50 subtasks to construct the test set. Specifically, we use the subtasks indexed 5–54 as the test set. We also created a validation set from subtasks indexed 55–69.

### C.1.4. SCIBENCH

SciBench is a benchmark designed to evaluate large language models on college-level scientific problem-solving tasks across mathematics, physics, and chemistry. Each task is open-ended and free-response, requiring multi-step reasoning, conceptual understanding, and the application of domain-specific knowledge.

The benchmark consists of 692 problems curated from widely-used undergraduate textbooks. The input for each task includes a textual problem statement, and the expected output is a correct, well-reasoned answer or solution. Tasks span a diverse set of scientific domains, including calculus, differential equations, electromagnetism, thermodynamics, and quantum chemistry.

We retain 15% of the questions from each domain as the validation set, and use the remaining 85% as the test set.

### C.2. Detailed Baseline Descriptions

#### C.2.1. GAME OF 24

- Input-Output (IO) prompting uses the LLM to directly generate an output, with no intermediate steps.

- Chain-of-Thought (CoT) (Wei et al., 2022) solves the problem step by step by decomposing it into a sequence of thoughts.

- Chain-of-Thought (Wei et al., 2022) with Self-Consistency (Wang et al., 2022) CoT-SC, generates multiple responses for the same CoT prompt and then selects the best one based on majority voting.

- Algorithm-of-Thoughts (AoT) (Sel et al., 2024) (AoT), guides the reasoning through algorithmic pathways by including such examples in its prompt.

- Tree-of-Thoughts (ToT) (Yao et al., 2024). decomposes the problem into multiple chain of thoughts, organized in a tree structure. Thought evaluation and search traversal algorithms are utilized to solve the problem.

- Graph of Thoughts (GoT) (Besta et al., 2024) allows the organization of thoughts in a graph structure. It introduces arbitrary graph-based thought transformations such as thought aggregation and thought refinement.

- Reason for futre, act for now (RAFA) (Liu et al., 2024) structures its reasoning by initially implementing a potential plan for a trajectory. Then, feedback is gathered for actions included in the plan. Finally, a new plan is generated with the gathered feedback in context. Even though we compared with RAFA using models GPT3.5 and Llama 3.2 11B, we did not compare with GPT4 or Llama 3.2 90b as its cost would be prohibitive.

- Reasoning via Planning (RAP) (Hao et al., 2023) augments LLMs with a world model and employs Monte Carlo Tree Search (MCTS)-based planning to generate and traverse its thought process. Language Agent Tree Search (LATS) (Zhou et al., 2024) extends this concept by leveraging environment interactions, thereby eliminating the need for a world model. We did not compare with any of these two methods as code for the game of 24 task was not available.

- The Buffer of Thoughts (BoT) (Yang et al., 2024) framework extracts task-specific information, uses it to retrieve relevant thought templates from its meta-buffer, and then instantiates them with more task-specific reasoning structures before continuing with the reasoning process. For BoT (Yang et al., 2024) we were unable to reproduce the results reported in their paper as a component of their method (meta-buffer) is not available and we have no detailed instructions on how to recreate it ourselves.

- The LLMCompiler (Kim et al., 2024) is an LLM compiler that optimizes the parallel function calling performance of LLMs. There was no code available for the task of Game of 24, so we do not compare against it.

- Tree of uncertain Thoughts (TouT) (Mo & Xin, 2024) leverages Monte Carlo Dropout to quantify uncertainty scores associated with LLMs' diverse local responses at intermediate thoughts. This local uncertainty quantification is then united with global search algorithms. There was no code available so we do not compare against TouT.

- ReST-MCTS* (Zhang et al., 2024) introduces a self-training framework that uses a modified Monte Carlo Tree Search (MCTS*) guided by process-level rewards. Instead of relying solely on final answers, it infers per-step rewards to identify and reinforce high-quality reasoning traces, improving the LLM's reasoning ability over successive iterations. In our experiments, we only tested the MCTS* component for inference-time reasoning, without the full iterative self-training loop, to ensure a fair comparison, as all other baselines were also evaluated purely in the inference-time setting.

### C.2.2. MINI CROSSWORDS

- Input-Output (IO) prompting uses the LLM to directly generate an output, with no intermediate steps.

- Chain-of-Thought (CoT) (Wei et al., 2022) solves the problem step by step by decomposing it into a sequence of thoughts.

- Chain-of-Thought (Wei et al., 2022) with Self-Conistency (Wang et al., 2022) CoT-SC, generates multiple responses for the same CoT prompt and then selects the best one based on majority voting.

- Algorithm-of-Thoughts (AoT) (Sel et al., 2024) (AoT), guides the reasoning through algorithmic pathways by including such examples in its prompt. For its Mini Crosswords implementation, AoT utilizes 2 prompts that need to be run sequentialy and provides both of them. However, in between the two prompts a necessary step is performed which extracts the word combination of the highest "compatibility". No further details are found about this step and since it can be interpreted in a number of ways we chose not to move on with it.

- Tree-of-Thoughts (ToT) (Yao et al., 2024). decomposes the problem into multiple chain of thoughts, organized in a tree structure. Thought evaluation and search traversal algorithms are utilized to solve the problem.

- Graph of Thoughts (GoT) (Besta et al., 2024) allows the organization of thoughts in a graph structure. It introduces arbitrary graph-based thought transformations such as thought aggregation and thought refinement.

### C.2.3. WEBSHOP

- Act (Yao et al., 2023) simply prompts the framework to perform an action within a closed loop.

- ReAct (Yao et al., 2023) integrates reasoning into Act by allowing the model to think instead of explicitly performing an action to the environment.

- Reflexion (Shinn et al., 2023) generates linguistic feedback that is utilized during subsequent runs.

- Agent with State-Space ExploRation (LASER) (Ma et al., 2023) models environment interactive tasks as state-space exploration. This is achieved by allowing the LLM agent to transition among a pre-defined set of states by performing actions to complete the task.

- Language Agent Tree Search (LATS) (Zhou et al., 2024) employs Monte Carlo Tree Search (MCTS)-based planning to generate and traverse its thought process, leveraging environment interactions. Even though we compare with LATS using GPT3.5, we do not repeat the experiment for any other model as it is prohibitively expensive.

- Multiple deep learning approaches(Yao et al., 2022). We also compare with a variety of deep learning approaches such as supervised, reinforcement and imitation learning. We report their average score as given in their published paper.

- Human Experts: A human annotators have been recruited by the Webshop authors (Yao et al., 2022) to study their trajectories. Based on their results, thirteen of them are recruited and trained further. Finally, the top 7 performers are selected as experts.(Yao et al., 2022)

- Retrieval-Augmented Planning (RAP) (Kagaya et al., 2024) dynamically leverage past experiences corresponding to the current situation and context in both textual and multimodal environments.

- The LLMCompiler (Kim et al., 2024) is an LLM compiler that optimizes the parallel function calling performance of LLMs. There was no code available for the task of WebShop, so we do not compare against it.

C.2.4. SCIBENCH

- Input-Output (IO) prompting uses the LLM to directly generate an output, with no intermediate steps.

- Chain-of-Thought (CoT) (Wei et al., 2022) solves the problem step by step by decomposing it into a sequence of thoughts.

- Chain-of-Thought (Wei et al., 2022) with Self-Conistency (Wang et al., 2022) CoT-SC, generates multiple responses for the same CoT prompt and then selects the best one based on majority voting.

- Tree-of-Thoughts (ToT) (Yao et al., 2024). decomposes the problem into multiple chain of thoughts, organized in a tree structure. Thought evaluation and search traversal algorithms are utilized to solve the problem.

- ReST-MCTS* (Zhang et al., 2024) introduces a self-training framework that uses a modified Monte Carlo Tree Search (MCTS*) guided by process-level rewards. Instead of relying solely on final answers, it infers per-step rewards to identify and reinforce high-quality reasoning traces, improving the LLM's reasoning ability over successive iterations. In our experiments, we only tested the MCTS* component for inference-time reasoning, without the full iterative self-training loop, to ensure a fair comparison, as all other baselines were also evaluated purely in the inference-time setting.

## C.3. Implementation Details

**Platforms.** GPT models were were accessed through the OpenAI API while the utilization of the Llama models was facilitated by the TogetherAI API.

**Model checkpoints and prices.** To compute the costs of our experiments we used the current model prices indicated OpenAI and Together AI, accordingly to the model. The specific models snapshot we used, along with their respective prices are presented in 5.

|  | US$ per 1m prompt tokens | US$ Per 1m completion tokens |
|---|---|---|
| **gpt-3.5-turbo-0125** | 0.5 | 1.5 |
| **gpt-4-0613** | 30.0 | 60.0 |
| **meta-llama/Llama-3.2-90B-Vision-Instruct-Turbo** | 1.2 | 0.06 |
| **meta-llama/Llama-3.2-11B-Vision-Instruct-Turbo** | 0.18 | 0.18 |

Table 5: **Model snapshot prices.** OpenAI and TogetherAI prices for each model used, during the implementation of the project.

**Model configurations.** Generation parameters specified when making calls to any of the models used throughout this project. These parameters were not defined by us, but by the implementation where the respective prompts where introduced. Specifically, Game of 24 and Mini Crosswords parameters were used from (Yao et al., 2024), WebShop step request parameters was taken from (Yao et al., 2023), WebShop evaluate request parameters from (Zhou et al., 2024) and SciBench parameters are taken from (Zhang et al., 2024). Configurations presented in Table 6.

|  | max_tokens | temperature | top_p | stop |
|---|---|---|---|---|
| **Game of 24** | 100 | 0.7 | 1 | null |
| **Mini Crosswords** | 1000 | 0.7 | 1 | null |
| **WebShop (step)** | 100 | 1 | 1 | ["\n"] |
| **WebShop (eval)** | 100 | 1 | 1 | null |
| **SciBench** | 1000 | 0.7 | 1 | null |

Table 6: **Generation parameters.** Generation parameters specified when making requests to any model.

**Base model selection strategy.** We selected GPT4 to be our base model as it was the one for which the prompts we used were originally designed for. Excepions were made for the RAFA (Liu et al., 2024) and LATS (Zhou et al., 2024) baselines

as their cost was prohibitive for us to run using GPT4. Additionally, for the task of WebShop (Yao et al., 2022), we did not repeat any baseline for GPT4. That was because firstly, the price was extremely steep for some of the baselines and secondly because some of the baselines had already achieved near-human level of performance. Finally, for the SciBench task (Wang et al., 2024a), we did not replicate any baselines for GPT-4 or Llama-3.2-90B due to the fact that SciBench spans 10 datasets across different domains, incurring prohibitively high computational costs.

**Prompts.** This section provides all the prompts used for the models evaluated in our experiments. We include the exact phrasing and formatting of each prompt to ensure reproducibility and allow for detailed examination of how the tasks were presented to the models.

```
Input: 2 8 8 14
Possible next steps:
2 + 8 = 10 (left: 8 10 14)
8 / 2 = 4 (left: 4 8 14)
14 + 2 = 16 (left: 8 8 16)
2 * 8 = 16 (left: 8 14 16)
8 - 2 = 6 (left: 6 8 14)
14 - 8 = 6 (left: 2 6 8)
14 /  2 = 7 (left: 7 8 8)
14 - 2 = 12 (left: 8 8 12)
Input: {input}
Possible next steps:
```

Prompt 1: **Game of 24 - Step prompt** The prompt used to generate candidate new states. Taken from (Yao et al., 2024).

```
Use numbers and basic arithmetic operations (+ - * /) to obtain 24. Each step, you are
    only allowed to choose two of the remaining numbers to obtain a new number.
Input: 4 4 6 8
Steps:
4 + 8 = 12 (left: 4 6 12)
6 - 4 = 2 (left: 2 12)
2 * 12 = 24 (left: 24)
Answer: (6 - 4) * (4 + 8) = 24
Input: 2 9 10 12
Steps:
12 * 2 = 24 (left: 9 10 24)
10 - 9 = 1 (left: 1 24)
24 * 1 = 24 (left: 24)
Answer: (12 * 2) * (10 - 9) = 24
Input: 4 9 10 13
Steps:
13 - 10 = 3 (left: 3 4 9)
9 - 3 = 6 (left: 4 6)
4 * 6 = 24 (left: 24)
Answer: 4 * (9 - (13 - 10)) = 24
Input: 1 4 8 8
Steps:
8 / 4 = 2 (left: 1 2 8)
1 + 2 = 3 (left: 3 8)
3 * 8 = 24 (left: 24)
Answer: (1 + 8 / 4) * 8 = 24
Input: 5 5 5 9
Steps:
5 + 5 = 10 (left: 5 9 10)
10 + 5 = 15 (left: 9 15)
15 + 9 = 24 (left: 24)
Answer: ((5 + 5) + 5) + 9 = 24
Input: {input}
```

Prompt 2: **Game of 24 - Last step prompt** In the game of 24, once all initial numbers were combined, if the resulting number is 24, then the following chain of thought prompt was used to summarized the operations that have taken place to get there (Yao et al., 2024).

```
Evaluate if given numbers can reach 24 (sure/likely/impossible)
10 14
10 + 14 = 24
sure
11 12
11 + 12 = 23
12 - 11 = 1
11 * 12 = 132
11 / 12 = 0.91
impossible
4 4 10
4 + 4 + 10 = 8 + 10 = 18
4 * 10 - 4 = 40 - 4 = 36
(10 - 4) * 4 = 6 * 4 = 24
sure
4 9 11
9 + 11 + 4 = 20 + 4 = 24
sure
5 7 8
5 + 7 + 8 = 12 + 8 = 20
(8 - 5) * 7 = 3 * 7 = 21
I cannot obtain 24 now, but numbers are within a reasonable range
likely
5 6 6
5 + 6 + 6 = 17
(6 - 5) * 6 = 1 * 6 = 6
I cannot obtain 24 now, but numbers are within a reasonable range
likely
10 10 11
10 + 10 + 11 = 31
(11 - 10) * 10 = 10
10 10 10 are all too big
impossible
1 3 3
1 * 3 * 3 = 9
(1 + 3) * 3 = 12
1 3 3 are all too small
impossible
{input}
```

Prompt 3: **Game of 24 - Value prompt** The prompt used to evaluate a state (Yao et al., 2024).

```
Let's play a 5 x 5 mini crossword, where each word should have exactly 5 letters.

{input}

Given the current status, list all possible answers for unfilled or changed words, and
    your confidence levels (certain/high/medium/low), using the format "h1. apple (medium)
    ". Use "certain" cautiously and only when you are 100% sure this is the correct word.
    You can list more then one possible answer for each word.
```

Prompt 4: **Mini Crosswords - Step prompt** The prompt used to generate candidate new states (Yao et al., 2024).

```
Evaluate if there exists a five letter word of some meaning that fit some letter
    constraints (sure/maybe/impossible).

Incorrect; to injure: w _ o _ g
The letter constraint is: 5 letters, letter 1 is w, letter 3 is o, letter 5 is g.
Some possible words that mean "Incorrect; to injure":
wrong (w r o n g): 5 letters, letter 1 is w, letter 3 is o, letter 5 is g. fit!
sure

A person with an all-consuming enthusiasm, such as for computers or anime: _ _ _ _ u
The letter constraint is: 5 letters, letter 5 is u.
```

```
Some possible words that mean "A person with an all-consuming enthusiasm, such as for
    computers or anime":
geek (g e e k): 4 letters, not 5
otaku (o t a k u): 5 letters, letter 5 is u
sure

Dewy; roscid: r _ _ _ l
The letter constraint is: 5 letters, letter 1 is r, letter 5 is l.
Some possible words that mean "Dewy; roscid":
moist (m o i s t): 5 letters, letter 1 is m, not r
humid (h u m i d): 5 letters, letter 1 is h, not r
I cannot think of any words now. Only 2 letters are constrained, it is still likely
maybe

A woodland: _ l _ d e
The letter constraint is: 5 letters, letter 2 is l, letter 4 is d, letter 5 is e.
Some possible words that mean "A woodland":
forest (f o r e s t): 6 letters, not 5
woods (w o o d s): 5 letters, letter 2 is o, not l
grove (g r o v e): 5 letters, letter 2 is r, not l
I cannot think of any words now. 3 letters are constrained, and _ l _ d e seems a common
    pattern
maybe

An inn: _ d _ w f
The letter constraint is: 5 letters, letter 2 is d, letter 4 is w, letter 5 is f.
Some possible words that mean "An inn":
hotel (h o t e l): 5 letters, letter 2 is o, not d
lodge (l o d g e): 5 letters, letter 2 is o, not d
I cannot think of any words now. 3 letters are constrained, and it is extremely unlikely
    to have a word with pattern _ d _ w f to mean "An inn"
impossible

Chance; a parasitic worm; a fish: w r a k _
The letter constraint is: 5 letters, letter 1 is w, letter 2 is r, letter 3 is a, letter 4
    is k.
Some possible words that mean "Chance; a parasitic worm; a fish":
fluke (f l u k e): 5 letters, letter 1 is f, not w
I cannot think of any words now. 4 letters are constrained, and it is extremely unlikely
    to have a word with pattern w r a k _ to mean "Chance; a parasitic worm; a fish"
impossible

{input}
```

Prompt 5: **Mini Crosswords - Value prompt** The prompt used to evaluate a state. Specifically, this prompt evaluates a potential solution of 1 out of the 10 questions of the Crossword. To get the value of the overall state this prompt was called for each row/column of the crosswords board with a potential solution (Yao et al., 2024).

```
Webshop
Instruction:
i would like a 3 ounce bottle of bright citrus deodorant for sensitive skin, and price
    lower than 50.00 dollars
[Search]

Action: search[3 ounce bright citrus deodorant sensitive skin]
Observation:
[Back to Search]
Page 1 (Total results: 50)
[Next >]
[B078GWRC1J]
Bright Citrus Deodorant by Earth Mama | Natural and Safe for Sensitive Skin, Pregnancy and
    Breastfeeding, Contains Organic Calendula 3-Ounce
$10.99
[B078GTKVXY]
```

```
Ginger Fresh Deodorant by Earth Mama | Natural and Safe for Sensitive Skin, Pregnancy and
    Breastfeeding, Contains Organic Calendula 3-Ounce
$10.99
[B08KBVJ4XN]
Barrel and Oak - Aluminum-Free Deodorant, Deodorant for Men, Essential Oil-Based Scent,
    24-Hour Odor Protection, Cedar & Patchouli Blend, Gentle on Sensitive Skin (Mountain
    Sage, 2.7 oz, 2-Pack)
$15.95

Action: think[B078GWRC1J and B078GTKVXY are bright citrus deodorant less then 50 dollars.
    I can check B078GWRC1J first.]
Observation: OK.

Action: click[B078GWRC1J]
Observation:
[Back to Search]
[< Prev]
scent [assorted scents][bright citrus][calming lavender][ginger fresh][simply non-scents]
size [travel set (4-pack)][3 ounce (pack of 1)][3-ounce (2-pack)]
Bright Citrus Deodorant by Earth Mama | Natural and Safe for Sensitive Skin, Pregnancy and
     Breastfeeding, Contains Organic Calendula 3-Ounce
Price: $10.99
Rating: N.A.
[Description]
[Features]
[Reviews]
[Buy Now]

Action: think[For 3 ounce bottle of bright citrus deodorant for sensitive skin, the item
    has options 'bright citrus' and '3 ounce (pack of 1)' and seems good to buy.]
Observation: OK.

Action: click[bright citrus]
Observation: You have clicked bright citrus.

Action: click[3 ounce (pack of 1)]
Observation: You have clicked 3 ounce (pack of 1).

Action: click[Buy Now]
```

Prompt 6: **WebShop - Step prompt** The prompt used to generate the next action so that after performing the action a new state would be achieved. The observation and action history of the agent would be appended at the end of this prompt. Taken from (Yao et al., 2023).

```
Given an item to purchase and a trajectory that aims to buy an item that exactly matches
    the specification, which corresponds to the ideal score of 1.0, analyze the following
    trajectory, then at the last line conclude "Thus the correctness score is {s}", where
    s is an integer from 1 to 10.

Here are some examples

Webshop
Instruction:
i am looking for dairy free and apple variety pack of chips, and price lower than 30.00
    dollars
[Search]

Action: search[dairy free and apple variety pack of chips]
Observation:
[Back to Search]
Page 1 (Total results: 50)
[Next >]
[B07HRFJWP8]
Enjoy Life Foods Soft Baked Ovals, Breakfast Bars, Nut Free Bars, Soy Free, Dairy Free,
    Non GMO, Gluten Free, Vegan, Variety Pack, 4 Boxes (20 Bars Total)
```

```
$100.0
[B01KMHY5PG]
Enjoy Life Soft Baked Chewy Bars, Variety Pack, Nut Free Bars, Soy Free, Dairy Free,
    Gluten Free, 6 Boxes (30 Total Bars)
$21.49
[B008D2X8C4]
Enjoy Life Lentil Chips Variety Pack, Dairy Free Chips, Soy Free, Nut Free, Non GMO, Vegan
    , Gluten Free, 24 Bags (0.8 oz)
$100.0

Action: think[The first two items are not what I want. The third item is the dairy free
    and apple variety pack of chips I want.]
Observation: OK.

Action: click[B008D2X8C4]
Observation:
[Back to Search]
[< Prev]
flavor name [dill and sour cream][garlic & parmesan][light sea salt][margherita pizza][
    thai chili lime][variety pack]
size [0.8 ounce (pack of 24)][4 ounce (pack of 12)]
Price: $100.0
Rating: N.A.
[Description]
[Features]
[Reviews]
[Buy Now]

Action: think[The item has options 'variety pack' and '0.8 ounce (pack of 24)' and seems
    good to buy.]
Observation: OK.

Action: click[variety pack]
Observation: You have clicked variety pack.

Action: click[0.8 ounce (pack of 24)]
Observation: You have clicked 0.8 ounce (pack of 24).

Action: click[Buy Now]

Reflection: In this attempt, I was unsuccessful. I accidentally bought a product that was
    $100, which is more than my budget of $30. Either way, the initial search results were
     not good. Next time, I will do search["variety pack of chips"] and then check if the
    results meet the dairy free and the $30 budget constraints. I will continue to refine
    my searches so that I can find more products.
Thus the correctness score is 5

Webshop
Instruction:
i would like a 3 ounce bottle of bright citrus deodorant for sensitive skin, and price
    lower than 50.00 dollars
[Search]

Action: search[3 ounce bright citrus deodorant sensitive skin]
Observation:
[Back to Search]
Page 1 (Total results: 50)
[Next >]
[B078GWRC1J]
Bright Citrus Deodorant by Earth Mama | Natural and Safe for Sensitive Skin, Pregnancy and
     Breastfeeding, Contains Organic Calendula 3-Ounce
$10.99
[B078GTKVXY]
Ginger Fresh Deodorant by Earth Mama | Natural and Safe for Sensitive Skin, Pregnancy and
    Breastfeeding, Contains Organic Calendula 3-Ounce
```

```
$10.99
[B08KBVJ4XN]
Barrel and Oak - Aluminum-Free Deodorant, Deodorant for Men, Essential Oil-Based Scent,
    24-Hour Odor Protection, Cedar & Patchouli Blend, Gentle on Sensitive Skin (Mountain
    Sage, 2.7 oz, 2-Pack)
$15.95

Action: think[B078GWRC1J and B078GTKVXY are bright citrus deodorant less then 50 dollars.
    I can check B078GWRC1J first.]
Observation: OK.

Reflection: Although the task is not yet complete, the first search is correct and returns
     relevant results. The thought is also logical and has potential to lead to the
    correct selection.
Thus the correctness score is 9
{input}
```

Prompt 7: **WebShop - Value prompt** The prompt used to evaluate a state. The observation and action history of the agent would be appended at the end of this prompt (Zhou et al., 2024).

```
Given a science problem, you need to answer the problem based on your existing knowledge.
    The input may include some existing steps to solve the question and you should
    continue to complete the solution based on these existing steps.

If the input does not provide any existing steps, you need give the first step in solving
    or calculating the problem. If partial solution steps are provided, you need to output
     the next step along the lines of the existing steps.
The output format is limited to: "Next step: ..." where ... indicates omitted output
    information, which is the next step in the answer that you should give. Your output
    must be a complete step, which may include detailed calculations, reasoning, choosing
    answers, etc. but no reasoning.

If the existing steps are already sufficient, you can output "The final answer is: $...$"
    where ... indicates the final answer to the question.

Please provide MULTIPLE alternative next steps. Use the following format:
"Next step: $...$
Next step: $...$
Next step: $...$".

Below is the input, please follow the specified format for your output.

Problem: {problem}
Existing steps:
{existing_steps}
Output:
```

Prompt 8: **SciBench - Step prompt** The prompt used to generate candidate new states. Taken from (Zhang et al., 2024)

```
Given a math problem and its corresponding solution, your task is to extract the final
    answer obtained in the solution.
You should summarize the answer using the format: "The final answer is $...$". Replace
    "..." with the answer obtained in the solution.
Problem: {problem}
Solution: {existing_steps}
Extracted answer:
```

Prompt 9: **SciBench - Summary prompt** This prompt is applied after a solution is found to adjust the output into the expected format. Taken from (Zhang et al., 2024)

```
Your task is to assess whether the provided solution steps can successfully solve the
    given science/mathematics problem and output a score.
The score should be a decimal between 0 and 1. If all the provided steps are incorrect (
    every step is wrong), the score should be 0. If all steps are correct and the final
```

```
      answer is successfully calculated, the score should be 1. The more errors there are in
      the steps, the closer the score should be to 0. The closer the steps are to the final
      correct answer, the closer the score should be to 1.
Steps that only contain verbal descriptions without any mathematical expressions should
    generally receive a low score. A score equal to or greater than 0.9 can only be given
    if the answer has already been calculated to a specific numerical value. If the
    thought process is complete but the answer is not computed, or only the mathematical
    expression is written without solving it, the score must be below 0.9.

First provide an analysis, then the score. Your analysis and scoring should be entirely
    based on the given steps. Do not continue solving the problem. Please study the
    following examples.

{examples}

Below is a problem and the existing steps, with analysis and scoring. Be careful not to
    output the next steps in the analysis, and the scoring should be based entirely on the
     steps given in the input.
The output format is limited to: "Analysis:...\nScore:...", where ... indicates omitted
    output content, which is the part you need to fill in.

Input:
Problem: {problem}
Existing steps:
{existing_steps}
Output:
```

Prompt 10: **SciBench - Value prompt** This prompt used to evaluate a state. The original prompt was taken from (Zhang et al., 2024) but was translated from Chinese to English using Google Translate.

**Practical extensions in the FOA framework.**

- **Caching:** The caching mechanism is utilized during the evaluation phase of our method to enhance its efficiency. It operates by ensuring that a given state is evaluated only once by the language model. This is achieved through a temporary state-to-value map maintained for the duration of a single run of the algorithm. Consequently, only when an agent encounters a previously unseen state, the LLM evaluates it and stores it in the cache. However, if a different agent (or the same agent at a later step) revisits that state, the LLM does not re-evaluate it; instead, the value is retrieved from the state-to-value cache. In comparison to other baselines such as ToT (Yao et al., 2024) and LATS (Zhou et al., 2024), no additional caching is being performed.

- **Batching:** During the mutation or selection phase (depending on the task) prompts are callected by all agents. Once that happens, if duplicated prompts occur, instead of making several individual requests for the same prompt, we make a single request and ask for multiple outputs. Employing batching in this way, ensures competitive fairness to methods such as ToT which utilize the this mechanic in the same way. This approach enhances efficiency and resource management by reducing network latency, server requests, on top of lowering the costs, as the user pays for the input tokens only once. Additionally, batching ensures consistency, as slight changes in the model's state or data processing on the provider's side, can affect individual requests differently.

**Task-specific modifications.**

- For the Scibench task (Wang et al., 2024a), we used the prompts provided in the ReST-MCTS* paper (Zhang et al., 2024). However, from our understanding, the evaluation prompt used in that paper was written in Chinese, and no official English version was available. To ensure better control over the evaluation process and to align the task with the predominantly English-language setup of our experiments, we translated the original Chinese prompt into English. This allowed us to directly inspect, adjust, and verify the evaluation inputs, ensuring consistency and clarity across all baselines.

## C.4. Hyperparameter Tuning

We perform a hyperparameter grid-search on the validation set, to explore trade-offs between success rate and cost. For this search, we use GPT-3.5-turbo, since GPT-4 would be prohibitively expensive. The hyperparameters we consider are the number of agents, the total number of steps each agent is allowed to perform, the discount factor $\gamma$, the resampling frequency $k$ and the resampling method.

The grid search is implemented in two steps. Initially, a broader, more general grid search is conducted to obtain an approximate understanding of where the optimal configurations are located. Subsequently, a more precise grid search is performed based on the findings from the initial step. The results of the second grid search for Game of 24 are presented in Figure 9, for Mini Crosswords in Figure 10 and for WebShop in Figure 11.

### C.4.1. GAME OF 24

The strategy for selecting the optimal configuration for the Game of 24 involves choosing the configuration that yields the best performance at the lowest cost. Following this approach, it is evident that the optimal number of agents and steps is achieved when the either hyperparameter is set to 9 or 12. However, since the cost is lowest at 9, this value is chosen for both the number of agents and the number of steps. Regarding the resampling frequency, resampling after every step (i.e., $k = 1$) results in significantly better performance and is therefore selected. For the discount factor $\gamma$, no notable differences in performance or cost are observed. Thus, $\gamma = 0.5$ is chosen as it represents a balanced choice between not allowing backtracking ($\gamma = 0$) and maximally encouraging backtracking by rendering the discount factor inconsequential ($\gamma = 1$). Finally, the linear filtered resampling method provides similar results to the linear method but at a significantly lower cost, making it the preferred choice.

The linear filtered resampling method is essentially the same as linear resampling, but it only considers states whose values are equal to or greater than the value of the current best-evaluated state. Across the different tasks, we found that this method is advantageous only when multiple states with sparse values are taken into consideration during resampling.

Table 7: Comparing FoA with previous using *accuracy* (↑ better) and *cost* (↓ better) on the detailed SciBench domains (base model: GPT-3.5). The best performance is shown in blue whereas the second best is shown in orange.

| Dataset | Accuracy (%) | | | | | | Cost (US$) | | | | | |
|---|---|---|---|---|---|---|---|---|---|---|---|---|
| | IO | CoT | CoT-SC | ToT | ReST-MCTS* | FoA | IO | CoT | CoT-SC | ToT | ReST-MCTS* | FoA |
| atkins | 1.1 | 7.6 | 8.1 | 11.1 | **15.5** | 14.5 | 0.01 | 0.07 | 1.37 | 18.35 | 9.41 | 1.54 |
| calculus | 4.6 | 0.4 | 0.3 | 3.4 | **14.1** | 10.2 | 0.01 | 0.02 | 0.45 | 6.00 | 3.78 | 0.52 |
| chemmc | 16.8 | 14.2 | 14.3 | 10.2 | 15.4 | **17.5** | 0.01 | 0.02 | 0.44 | 6.81 | 1.22 | 0.51 |
| class | 0.3 | 12.3 | 12.6 | 2.1 | 3.7 | 5.8 | 0.01 | 0.03 | 0.31 | 7.97 | 4.94 | 0.63 |
| diff | 4.9 | 1.0 | 3.5 | 7.9 | 12.1 | **14.3** | 0.01 | 0.03 | 0.58 | 9.12 | 8.54 | 0.67 |
| fund | 8.3 | 7.3 | 7.1 | 8.9 | **15.4** | 13.3 | 0.01 | 0.04 | 0.76 | 11.98 | 5.77 | 1.02 |
| matter | 2.4 | 9.4 | 10.3 | 3.7 | 6.1 | 8.1 | 0.01 | 0.03 | 0.69 | 10.33 | 5.43 | 0.77 |
| quan | 5.1 | 7.2 | 5.9 | 5.9 | 7.5 | **9.7** | 0.01 | 0.02 | 0.44 | 6.47 | 1.99 | 0.44 |
| stat | 13.7 | 21.1 | 20.3 | 16.4 | 3.3 | 16.4 | 0.01 | 0.04 | 0.68 | 12.65 | 3.78 | 0.9 |
| thermo | 1.8 | 3.4 | 4.8 | 0 | 0 | 0 | 0.01 | 0.04 | 0.65 | 13.68 | 6.21 | 0.98 |
| Average | 5.9 | 8.4 | 8.7 | 7.2 | 9.3 | **11.0** | 0.01 | 0.03 | 0.64 | 10.34 | 5.11 | 0.80 |

Game of 24 - Hyperparameter tuning

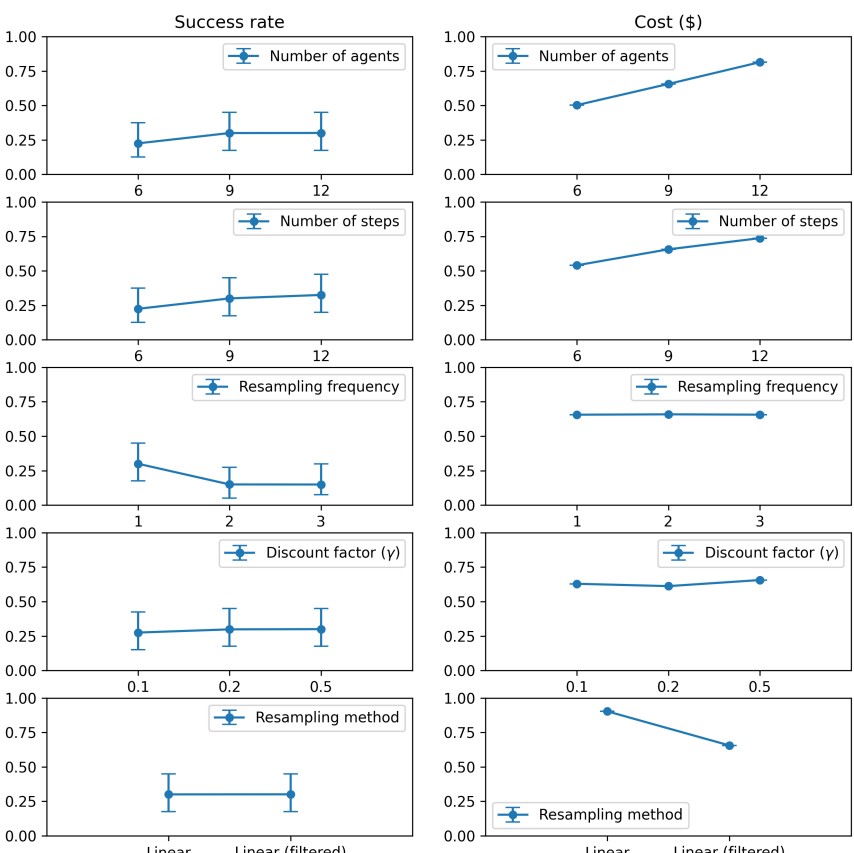

Figure 9: **Results of the final grid-search for Game of 24.** Each subplot illustrates the performance (left) and cost (right) as a function of a varying hyperparameter. The values of the remaining hyperparameters are set to those of the final, optimal configuration.

### C.4.2. MINI CROSSWORDS

For the Mini Crosswords task, in our final grids-search we observed that performance plateaued at approximately 0.4 overlap percentage, with minimal variation. Consequently, our primary strategy for this task was to minimize cost. The overlap remained similar when tuning the number of agents, number of steps, and the resampling frequency. However, in each case, there was always a specific value that significantly minimized cost, and this value was selected. Thus, we opted for 2 agents, running for 6 steps each, and resampling every $k = 3$ steps. For the discount factor and the resampling method, there was no clear advantage in terms of performance or cost. Therefore, we selected the most moderate options in each category: a discount factor of $\gamma = 0.5$ and the linear resampling method.

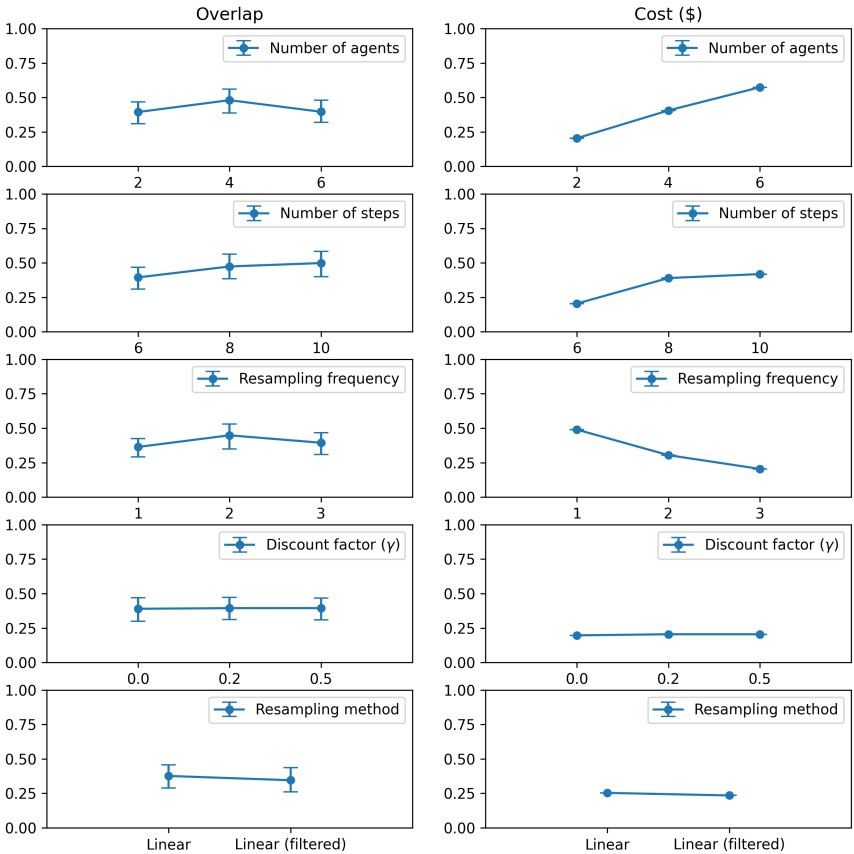

Figure 10: **Results of the final grid-search for Mini Crosswords.** Each subplot illustrates the performance (left) and cost (right) as a function of a varying hyperparameter. The values of the remaining hyperparameters are set to those of the final, optimal configuration.

### C.4.3. WEBSHOP

Finally, for the WebShop task, we reverted to our original strategy: achieving the best performance at the lowest possible cost. It is noteworthy that, due to the complexity of the WebShop task, more agents and steps were required for optimal performance. Consequently, we tested a broader range of values for both resampling frequency and discount factor compared to the previous tasks. The results indicated that the best average scores at the lowest cost were achieved with 15 agents running for 10 steps each. For the resampling frequency, the best scores were obtained with $\gamma \in \{2, 4, 5\}$, with 4 and 5 being the most cost-effective. Since there was no significant cost difference between 4 and 5, we selected 4 to allow for more frequent resampling. Finally, we omitted filtering during resampling as it provided no additional advantage.

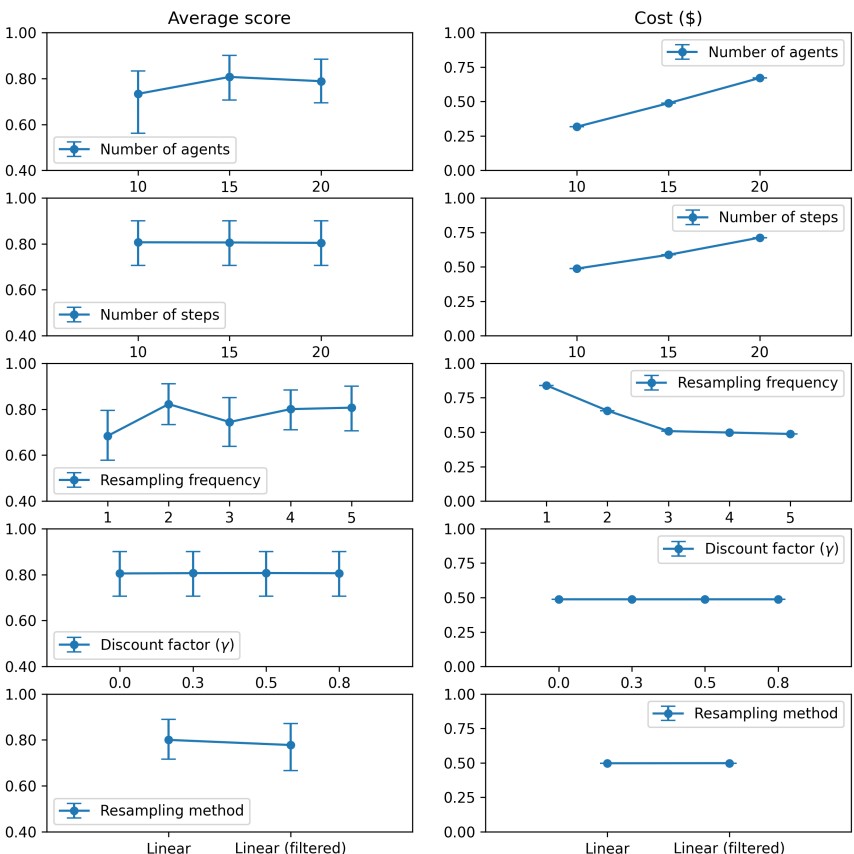

Figure 11: **Results of the final grid-search for WebShop.** Each subplot illustrates the performance (left) and cost (right) as a function of a varying hyperparameter. The values of the remaining hyperparameters are set to those of the final, optimal configuration.

# D. Additional Results

## D.1. Generalizability of the findings with other base models.

In the following section, we present our results for various models and demonstrate that the findings generalize across different settings. You can find the results for the task of Game of 24 in Table 8, Mini Crosswords in Table 9, WebShop in Table 10, and SciBench in Table 11.

**Note on the performance of Act and ReAct.** Act and ReAct have essentially the same architecture in the sense that an initial prompt is repeatedly being given to the LLM while it's being updated by the actions that have been chosen and the resulting environment observations. The only difference is that the ReAct prompt introduces the possibility of a new action for the LLM : "Think". When this action is chosen the LLM does not interact with the environment, it simply states its thoughts and aims to come up with a strategy to solve the problem. However, these prompts came up years ago where much less powerful models were used. As a result, more contemporary models interact differently with them.

Table 8: Comparing FOA with previous methods using *success rate* (↑ better) and *cost* (↓ better) on the Game of 24 task. The best and second best performances are shown in **blue** and **orange**, respectively. Owing to its exorbitant cost (≃ 600 US$), we could not run RAFA (shown as DNR).

| | **Task : Game of 24** | | |
|---|---|---|---|
| Model | Method | Success Rate (%) | Cost (US$) |
| GPT-3.5 | IO | 6.8 | **0.05** |
| | CoT | 3.6 | **0.12** |
| | CoT-SC | 5.2 | 1.40 |
| | AoT | 1.0 | 0.32 |
| | ToT | **13.6** | 1.71 |
| | GoT | 11.2 | 1.60 |
| | RAFA | 8.0 | 10.47 |
| | FoA | **25.1** | 1.55 |
| GPT-4 | IO | 6.0 | **0.65** |
| | CoT | 6.0 | **6.98** |
| | CoT-SC | 10.0 | 49.39 |
| | AoT | 49.0 | 20.98 |
| | ToT | **74.0** | 75.02 |
| | GoT | 63.0 | 70.00 |
| | RAFA | DNR | DNR |
| | FoA | **76.0** | 62.93 |
| Llama 3.2-11B | IO | 2.6 | **0.01** |
| | CoT | 3.6 | **0.01** |
| | CoT-SC | 4.8 | 0.05 |
| | AoT | **5.1** | 0.12 |
| | ToT | 2.7 | 0.37 |
| | GoT | 1.4 | 0.31 |
| | RAFA | 0.0 | 23.10 |
| | FoA | **6.0** | 0.32 |
| Llama 3.2-90B | IO | 6.0 | **0.03** |
| | CoT | 6.8 | **0.05** |
| | CoT-SC | 8.0 | 0.33 |
| | AoT | **36.8** | 0.81 |
| | ToT | 35.5 | 2.51 |
| | GoT | 30.1 | 2.11 |
| | RAFA | DNR | DNR |
| | FoA | **39.7** | 2.05 |

### D.2. Ablation analysis for the remaining tasks

In the following section, we present the remaining ablation studies we performed and display that our findings generalize across different settings. You can find the results for the Mini Crosswords ablation in Figure 12, for WebShop in Figure 13 and for SciBench in Figure 14.

### D.3. Details on RAFA Results and Metric Differences

In our evaluation of RAFA (Liu et al., 2024) for the Game of 24 task, we used the official implementation provided by the authors. However, the results we report differ from those presented in the original RAFA paper. This difference stems from a variation in the definition of the success rate evaluation metric.

Table 9: Comparing FOA with previous methods using *overlap* (↑ better) and *cost* (↓ better) on the Crosswords task. The best performance is shown in `blue` whereas the second best is shown in `orange`.

| Model | Method | Overlap (%) | Cost (US$) |
|---|---|---|---|
| **Task : Mini Crosswords** | | | |
| | IO | 31.2 | **0.01** |
| | CoT | 33.2 | **0.02** |
| | CoT-SC | 33.1 | 0.06 |
| GPT-3.5 | ToT | 33.3 | 0.48 |
| | GoT | **34.5** | 0.40 |
| | FoA | **36.2** | 0.25 |
| | IO | 36.8 | **0.51** |
| | CoT | 39.4 | **1.06** |
| GPT-4 | CoT-SC | 39.4 | 2.82 |
| | ToT | 39.7 | 49.99 |
| | GoT | **41.2** | 30.28 |
| | FoA | **46.0** | 12.94 |
| | IO | 6.2 | **0.01** |
| | CoT | 21.0 | **0.01** |
| Llama 3.2-11B | CoT-SC | 21.0 | 0.04 |
| | ToT | **46.5** | 0.44 |
| | GoT | 41.5 | 0.57 |
| | FoA | **50.9** | 0.16 |
| | IO | 5.0 | **0.04** |
| | CoT | 30.6 | **0.04** |
| Llama 3.2-90B | CoT-SC | 30.6 | 0.13 |
| | ToT | **62.8** | 6.01 |
| | GoT | 62.5 | 4.72 |
| | FoA | **64.92** | 1.55 |

Specifically, the RAFA paper reports results using a relaxed version of the success rate metric (see Footnote 1, page 50, ICML'24 camera-ready), which differs from the stricter formulation used in other benchmarks. To ensure consistency and fairness across all methods evaluated in our study, we applied the success rate implementation as defined in the original ToT (Yao et al., 2024) paper uniformly across all baselines.

As shown in Table 12, when we apply the relaxed success rate metric used in the original RAFA paper, our results align closely with those reported by its authors. This adjustment ensures that readers can understand the basis of any apparent discrepancies and interpret our comparisons across methods accurately.

Table 10: Comparing FOA with previous methods using *average score* (↑ better) and *cost* (↓ better) on the WebShop task. The best performance is shown in `blue` whereas the second best is shown in `orange`.

| | **Task : WebShop** | | |
| --- | --- | --- | --- |
| Model | Method | Average score | Cost (US$) |
| GPT-3.5 | Act | 58.1 | **0.10** |
| | ReAct | 48.7 | **0.17** |
| | Reflexion | 56.3 | 0.65 |
| | LASER | 57.2 | 0.41 |
| | LATS | **66.1** | 232.27 |
| | FoA | **75.6** | 1.68 |
| Llama 3.2-11B | Act | 28.2 | **0.10** |
| | ReAct | 16.7 | **0.12** |
| | Reflexion | 24.8 | 0.50 |
| | LASER | **54.0** | 0.75 |
| | LATS | - | - |
| | FoA | **77.2** | 2.60 |
| DL | IL (Yao et al., 2022) | 59.9 | - |
| | IL+RL (Yao et al., 2022) | **62.4** | - |
| | WebN-T5 (Gur et al., 2023) | 61.0 | - |
| | WebGUM (Furuta et al., 2024) | **67.5** | - |
| Human experts (Yao et al., 2022) | | 82.1 | - |

Table 11: Comparing FOA with previous methods using *accuracy* (↑ better) and *cost* (↓ better) on the SciBench task. The best performance is shown in `blue` whereas the second best is shown in `orange`.

| | **Task : SciBench** | | |
| --- | --- | --- | --- |
| Model | Method | Accuracy (%) | Cost (US$) |
| GPT-3.5 | IO | 5.9 | **0.01** |
| | CoT | 8.4 | **0.03** |
| | CoT-SC | 8.7 | 0.64 |
| | ToT | 7.2 | 10.34 |
| | ReST-MCTS* | **9.3** | 5.11 |
| | FoA | **11.0** | 0.80 |
| Llama 3.2-11B | IO | 0.9 | **0.02** |
| | CoT | 1.6 | **0.05** |
| | CoT-SC | 2.4 | 0.11 |
| | ToT | 1.9 | 2.35 |
| | ReST-MCTS* | **4.2** | 2.04 |
| | FoA | **5.1** | 0.27 |

| **RAFA results (Game-of-24)** | **Accuracy (%)** | **Low interval** | **High interval** |
| --- | --- | --- | --- |
| Our run with RAFA metric | 26 | 23 | 28 |
| Our run with ToT metric | 8 | 6 | 9 |
| RAFA run with RAFA metric | 29 | – | – |

Table 12: Comparison of RAFA results across different runs and evaluation metrics.

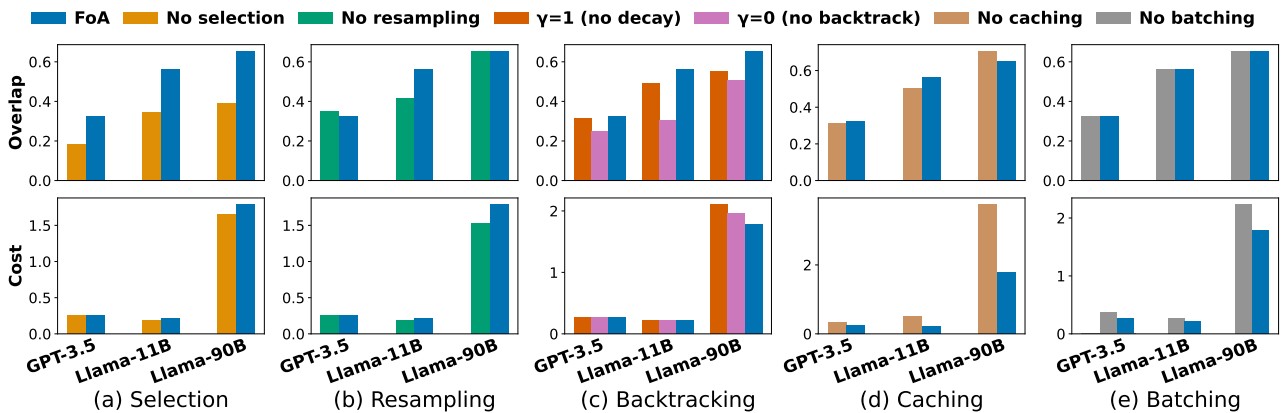

Figure 12: Ablation analysis to study the impact of (a) Selection phase, (b) Resampling, (c) Backtracking, (d) Caching, and (e) Batching on the performance of FOA using the Mini Crosswords task with GPT-3.5, Llama3.2-11B, and 90B as base models.

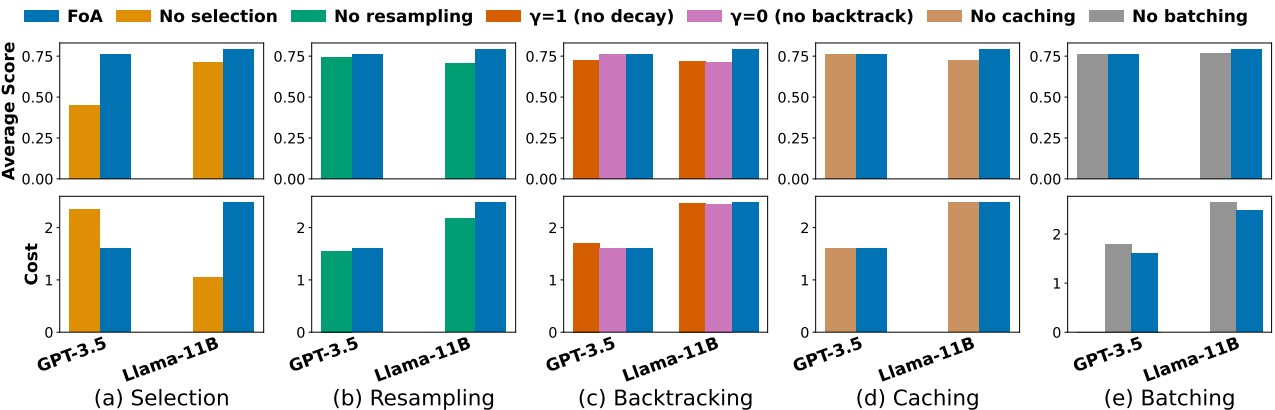

Figure 13: Ablation analysis to study the impact of (a) Selection phase, (b) Resampling, (c) Backtracking, (d) Caching, and (e) Batching on the performance of FOA using the WebShop task with GPT-3.5 and Llama3.2-11B base models.

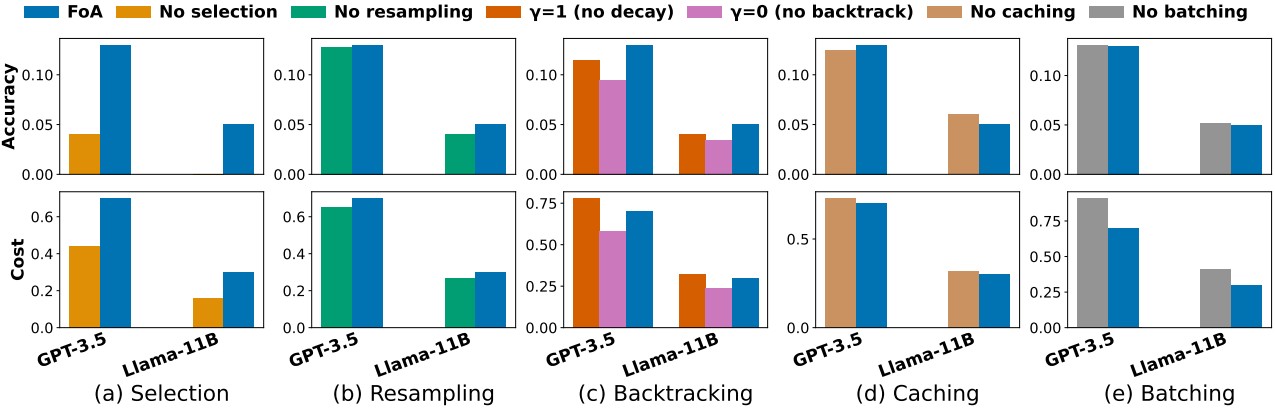

Figure 14: Ablation analysis to study the impact of (a) Selection phase, (b) Resampling, (c) Backtracking, (d) Caching, and (e) Batching on the performance of FOA using the SciBench task with GPT-3.5 and Llama3.2-11B base models.

