# OpenReview forum: "Fleet of Agents: Coordinated Problem Solving with Large Language Models"
_ICML.cc/2025/Conference — ICML 2025 poster_

### Official Review · Reviewer_Mfui · 2025-03-03

**Overall Recommendation:** 3

**Summary:**

The word produces a novel framework, FOA, that employes LLMs agents for dynamic tree search using genetic-type particle filtering approach. The multiple agents provide dynamic branching and adapting the exploration strategy. A major claim is to improve the cost-efficiency of multi-query methods. The paper compares FOA against several benchmarks like IO prompting, CoT, AoT, ToT, GoT, over tasks like Game of 24, Mini Crosswords, WebShop",with favorable results.

**Claims And Evidence:**

The claims are fairly evident as the method is developed through thr paper. They are
. FOA i

**Essential References Not Discussed:**

I'm not well positioned to enunciate that.

**Experimental Designs Or Analyses:**

This is a design oriented work. But it justifies the choices made with bnchmarking, ablation studies, and cost-quality trade-off and model size and quality analyses.

**Methods And Evaluation Criteria:**

The benchmark tasks appear appropriate because they require different reasoning, planning, and problem-solving skills. Using success rate (for "Game of 24"), overlap (for "Mini Crosswords"), and average score (for "WebShop") appears to be standard, judging from benchmark results being there.

**Other Comments Or Suggestions:**

-

**Other Strengths And Weaknesses:**

It is an architectural, engineering work. Unless the culture at icml is shifting, that is peripheral to   the former.

**Questions For Authors:**

-

**Relation To Broader Scientific Literature:**

Figure 2 IMV is a broad juxtaposition.

Unless the LR is insufficient, I see a good amount of positioning arounf *-of-thought reasoning models.

**Theoretical Claims:**

There aren't any.

---

> ### Author Rebuttal · Authors · 2025-04-01
>
> We thank the reviewer for their careful and thoughtful review. We appreciate the positive assessment, recognizing the new state-of-the-art performance of our method and its extensive evaluation. We are grateful for the reviewer’s thoughtful engagement, and hope that our clarifications help address the concerns raised. We will be happy to answer any further questions that the reviewer may have and would appreciate it if the reviewer would consider updating their score in light of this response.
>
> **Is this mainly an engineering work, and as such, a good fit for ICML?** We believe that the reviewer’s main concern originates from a misunderstanding about the nature of our work, which we clarify below with a point-by-point response.
>
> 1. **FoA is a principled algorithm.** We completely agree with the reviewer that our research is built on a significant engineering effort. However, our work also makes a meaningful contribution beyond architectural or engineering work. To the best of our knowledge, the application of genetic particle filtering to LLM-based reasoning processes is novel. We present a principled formulation of the algorithm and provide a deep empirical analysis of its behavior through extensive ablations.
>
> 2. **FoA is a runtime and doesn’t rely on crafted prompts.** We would like to highlight that unlike all existing works, we do not craft custom prompts for FOA, but instead, we reuse the prompts (cf. Appx. C.3 for details) provided by ToT (Yao et al., 2024) for the “Game of 24” and “Mini Crosswords” tasks and LATS (Zhou et al., 2024) for the “WebShop” task. This ensures a direct and fair comparison of the reasoning abilities of the benchmarked frameworks and controls for the impact of prompt quality, which is a confounder. (Lines 240-247, paper). Moreover, we would also like to highlight our response to **Reviewer JAyM** under **Comparison with ReST-MCTS\* [NeurIPS’24]**, where FoA was extended to a new benchmark task, SciBench, without any prompt tuning, **corroborating its strength as an algorithm**.
>
> 3. **ICML's historical inclination towards Applied ML research.** Finally, we believe our work aligns well with ICML’s long-standing interest in applied machine learning research, particularly work that enables new capabilities, systematically explores trade-offs, and provides reproducible insights grounded in empirical evaluation. Few examples below:
>     1. Fast support vector machine training and classification on graphics processors (ICML 2008) https://dl.acm.org/doi/10.1145/1390156.1390170
>     2. Making a Science of Model Search: Hyperparameter Optimization in Hundreds of Dimensions for Vision Architectures (ICML 2013)https://proceedings.mlr.press/v28/bergstra13.html
>     3. Scaling Laws for Fine-Grained Mixture of Experts (ICML 2024) https://proceedings.mlr.press/v235/ludziejewski24a.html
>
> **Access to supplementary material**
>
> All artifacts of our research, including code, data, and prompts, are publicly available at: https://anonymous.4open.science/r/FoA-4D83. Our results are fully reproducible.
>
> **Note:** Humbly and politely, it is our impression that the reviewer's comment in **Claims and Evidence** is cut-off. We welcome any feedback coming our way. If the reviewer would clarify their intention, we would greatly appreciate it and act upon it.

---

> > ### Comment · Reviewer_Mfui · 2025-04-06
> >
> > Agreeing with the argument.

---

> > > ### Author Response · Authors · 2025-04-07
> > >
> > > Dear Reviewer Mfui,
> > >
> > > Thanks a lot for taking the time to review our response and agreeing with the points outlined therein. Since the author feedback phase ends tomorrow, it would be great if you could kindly let us know if you require anything further from our side. Alternatively, we would sincerely appreciate it if you would consider updating your score in light of your agreement with our original response.
> > >
> > > Thanks once again for taking the time and writing a thoughtful review of our work. Appreciate it!
> > >
> > > Best,
> > > Paper 4052 authors

---

### Official Review · Reviewer_JAyM · 2025-03-14

**Overall Recommendation:** 3

**Summary:**

The paper proposes Fleet of Agents (FOA), a novel multi-agent framework leveraging genetic particle filtering to enhance problem-solving capabilities of Large Language Models (LLMs). FOA achieves improved reasoning quality and significantly reduces computational costs compared to state-of-the-art methods, demonstrating strong performance across various benchmark tasks and different sizes of language models.

**Claims And Evidence:**

The claims made in the submission are supported by clear and convincing evidence. Specifically, the authors present extensive experimental results across three diverse benchmark tasks ("Game of 24," "Mini-Crosswords," and "WebShop") using multiple LLMs (GPT-3.5, GPT-4, LLaMA3.2-11B, and LLaMA3.2-90B). Their primary claims regarding FOA’s effectiveness in improving reasoning quality (~5% increase) and substantially reducing computational costs (~60% decrease) compared to state-of-the-art methods are substantiated by detailed comparisons against multiple baselines (e.g., IO, CoT, CoT-SC, AoT, ToT, GoT, RAFA, LASER, LATS) on clearly defined metrics.

**Essential References Not Discussed:**

I would like to see comparison and discussion with ReST-MCTS

**Experimental Designs Or Analyses:**

Yes

 Why RAFA results don’t align with the paper “Reason for Future, Act for Now: A Principled Architectur for Autonomous LLM Agents” , what’s the difference between the origin codebase and your experiment

**Methods And Evaluation Criteria:**

Yes, the proposed methods and evaluation criteria are appropriate and make sense for the problem at hand.

**Other Comments Or Suggestions:**

1. In table 1, if you could not run RAFA, maybe just don’t present it.

**Other Strengths And Weaknesses:**

Strengths:

1. **Innovative Method** – The paper introduces a novel approach that advances the current state of the field.
2. **State-of-the-Art Results** – The proposed method achieves competitive or superior performance compared to existing benchmarks.
3. **Cost-Effective Solution** – The approach demonstrates strong performance while maintaining efficiency in terms of computational or resource costs.
Weaknesses:

**Clarity in Writing** – Some parts of the manuscript, particularly the mathematical formulations, lack clarity and could benefit from improved explanations and notation consistency.

**Questions For Authors:**

1. In line 166, how do u define an invalid state, , and how do the resembling of it uniform, it’s not quiet Clear. And since the bullet point here is sudden death, but you don’t mention what’s actually is define as sudden death which is ill writing

**Relation To Broader Scientific Literature:**

Related to test-time scaling method for LLM reasoning such as COT, TOT

**Theoretical Claims:**

1. In line188, right column, ‘, cat. resampling distr.’ are those typos? Could you make this mathematical part clearer, and also ‘iid’ maybe you should specify those preliminaries  in your articles? I mean while use those terminology, maybe also write those in lines, to better understand

2.  What do you want to express in line 194, right column

---

> ### Author Rebuttal · Authors · 2025-04-01
>
> We thank the reviewer for their thoughtful feedback. We are encouraged by their overall positive assessment, recognizing the novelty, strong performance, and cost-effectiveness of our method, and the extensive nature of our experiments involving diverse benchmark tasks, multiple baselines, and LLMs of different sizes. We hope to have comprehensively clarified all the concerns of the reviewer with our responses below. We will be happy to answer any further questions that the reviewer may have and hope that the reviewer considers adapting the overall assessment of our paper.
>
> **Comparison with ReST-MCTS\* [NeurIPS’24]**: We thank the reviewer for this suggestion. So far, we were able to perform the following new experiments, which, have only strengthened the paper further. Based on the reviewer’s feedback, we are willing to extend ReST-MCTS* to the remaining two tasks within the author feedback period. All new implementations are in our GitHub repository linked from the paper.
>
> 1. **ReST-MCTS\* on SciBench**: To ensure a correct understanding of their codebase and setup, we run MCTS* (as shown in the README) for the SciBench benchmark. Not only were we able to reproduce their results (with minor variations due to inherent LLM stochasticity) but also obtained prompts for SciBench, which allowed us to extend FoA to this new benchmark with minimal added efforts. [Table 4](https://pasteboard.co/yZGwwm3MNgY6.png) shows that on average, **FoA outperforms ReST-MCTS\* by obtaining a quality improvement of ~2% while reducing the cost by ~10%**. Once again, we’d like to highlight this point for all reviewers: **we do not craft custom prompts for FoA, but instead, we reuse the prompts provided in the official implementations, corroborating the strength of FoA as an algorithm**.
>
> 2. **ReST-MCTS\* on Game of 24**: With our improved understanding of the ReST-MCTS* codebase, we extended it to our Game of 24 benchmark.
>     1. **Tuning:** Similar to other benchmarked methods in our paper, we tuned the ReST-MCTS* hyperparameters using GPT-3.5 as the base model. We found that the branch parameter $b=5$ and iteration number $T=50$ led to the best success rate (cf. results [tuning b](https://pasteboard.co/GWbNwKHeq2sr.png) and [tuning T](https://pasteboard.co/GfuTCe4zvJqC.png)).
>     2. **Main results:** [Table 1](https://pasteboard.co/evCa1SRTigqo.png) presents results with GPT-4 as the base model and shows that FoA substantially outperforms ReST-MCTS* by being **\~40% better in quality while requiring only \~50% of the cost**. Owing to its exorbitant cost (~300 USD), we could not run ReST-MCTS* with $T=50$, and the presented results instead use $T=10$, thereby providing ReST-MCTS* with a similar cost budget as other baselines.
>
> **RAFA results**: We thank the reviewer for bringing this point up. We used the official implementation of RAFA provided by the authors. The difference in the results reported in our and the original RAFA paper stems from a **difference in the definition of the success rate evaluation metric**. RAFA used a relaxed version (cf. footer 1, page 50, ICML’24 camera-ready) of the success rate metric. However, for a fair comparison across all the benchmarked methods (ToT, GoT, RAFA, and FoA), we used the original implementation introduced in the ToT paper for all the baselines. As shown in [Table 5](https://pasteboard.co/5LIOBtxZKdmc.png), **using the relaxed success rate metric, our results align well** with those reported in the original RAFA paper.
>
> **Clarity in writing**: We thank the reviewer for their suggestions regarding writing clarity. Following is our point-by-point response to answer their concerns. We will utilize the extra page available for the final version to weave these details more cleanly into the main text.
> 1. **Firming up mathematical formulations.**  In our theoretical discussion (lines 185-194), we describe a resampling mechanism with replacement, where samples are drawn i.i.d. from a categorical distribution. We completely agree with the reviewer’s suggestion and will fix the equations by avoiding abbreviations in the notation, including terms such as ‘i.i.d.’ in preliminaries, and providing textual explanations of the equations.
> 2. **Invalid states and Sudden Death.** Depending on the task, some solution attempts may result in invalid states. For example, in a crossword puzzle, an agent might propose a word with the wrong number of characters, leading to an ill-defined state. It is standard practice, across algorithms (e.g. ToT, GoT), to prevent the search from continuing in such cases. For FoA, we use a simple filtering mechanism: when an agent enters an invalid state, it is deleted, and another agent is randomly duplicated to fill the gap. This is what we refer to as sudden death. The replacement agent is chosen uniformly at random, avoiding additional calls to the value function. We will include the aforementioned short description in Sec. 3.2 of the paper in the final version.

---

### Official Review · Reviewer_RBUc · 2025-03-14

**Overall Recommendation:** 3

**Summary:**

The paper introduces Fleet of Agents , a framework that coordinates multiple LLM agents using a genetic-type particle filtering approach to optimize the balance between exploration and exploitation in problem-solving tasks.

**Claims And Evidence:**

Yes

**Essential References Not Discussed:**

No

**Experimental Designs Or Analyses:**

Yes

**Methods And Evaluation Criteria:**

Yes

**Other Comments Or Suggestions:**

No

**Other Strengths And Weaknesses:**

Weakness:
1. I believe the three tasks are easy and a little bit too old.
2. Current FOA implementation uses a fixed number of agents per task, while an adaptive mechanism could improve efficiency.

**Questions For Authors:**

No

**Relation To Broader Scientific Literature:**

Good

**Theoretical Claims:**

Yes

---

> ### Author Rebuttal · Authors · 2025-04-01
>
> We thank the reviewer for their feedback and overall positive assessment. We hope to have comprehensively clarified all the questions and concerns of the reviewer with our responses below. We will be happy to answer any further questions that the reviewer may have and hope that the reviewer considers adapting the overall assessment of our paper.
>
> **Are the evaluation tasks adequate?** We thank the reviewer for bringing this point up. Following is our point-by-point response to answer their concern.
>
> 1. **Vetted benchmarks:** We believe that we have chosen a **judicious mix** of tasks. It is important that these tasks are well established in the literature, primarily to enable meaningful comparisons with prior work. Moreover, a recent [survey on LLM agents]((https://paperswithcode.com/paper/survey-on-evaluation-of-llm-based-agents#datasets)) highlights that our tasks are among the most commonly used in the field and remain representative of current evaluation standards, a point also recognized by the other two reviewers.
>
> 2. **Tasks appear easy.** While the tasks may appear easy from their description, they continue to pose a significant **challenge**, even to powerful proprietary methods. For example, using a carefully designed **CoT prompt, GPT-4 solves only 6%** of the GameOf24 challenges (our first task). Moreover, as portrayed by additional evaluation of open-source models in the appendix, our choice of tasks poses a **significant challenge for these smaller models**. For example, as shown in Table 8 in the appendix, while most methods achieve a high "average score" (>0.5) using GPT-3.5, only FoA portrays strong performance (average score ~0.75) with Llama 3.2-11B, whereas other methods struggle. This result corroborates the difficulty of the tasks and the strength of our FoA framework to work well with small open-source models.
>
> 3. **New task.** Finally, given the suggestion from **Reviewer JAyM**, we were able to successfully add results (cf. our response to **Reviewer JAyM** under **Comparison with ReST-MCTS\* [NeurIPS’24]**) on a new benchmark task, **SciBench**. [Table 4](https://pasteboard.co/yZGwwm3MNgY6.png) shows that on average, FoA outperforms ReST-MCTS* by obtaining a quality improvement of \~2% while reducing the cost by \~10%.
>
> **Would it be possible to use a dynamic number of agents in an adaptive fleet-size mechanism?**
> We completely agree with the reviewer that this is an interesting question and a great idea. In fact, we already highlighted this point under the limitations section (Section 7.3) of our paper. In the same section, we also bring up the concept of an adaptive fleet size as future work.
>
> Overall, we hope that our answers have satisfied the reviewer. We are confident in the strength of our contributions and believe that the experimental setup and results amply demonstrate the relevance and rigor of our work.

---

> > ### Comment · Reviewer_RBUc · 2025-04-07
> >
> > No further questions. I will keep my score.

---

> > > ### Author Response · Authors · 2025-04-09
> > >
> > > Dear Reviewer RBUc,
> > >
> > > Thanks, for your time and consideration!
> > >
> > > Best,
> > > Paper 4052 authors

---

### Decision · Program_Chairs · 2025-05-01

**Decision:**

Accept (poster)

**Comment:**

The reviewers reached an agreement on weak accept. Hope the authors can address the weaknesses in the camera-ready version.